# Finding Discriminative Filters for Specific Degradations in Blind Super-Resolution

Liangbin Xie[*1,2,3]    Xintao Wang[*3]    Chao Dong[1,4]    Zhongang Qi[3]    Ying Shan[3]

[1]Shenzhen Key Lab of Computer Vision and Pattern Recognition,
Shenzhen Institute of Advanced Technology, Chinese Academy of Sciences
[2]University of Chinese Academy of Sciences
[3]ARC Lab, Tencent PCG
[4]Shanghai AI Laboratory, Shanghai, China
{lb.xie, chao.dong}@siat.ac.cn
{xintaowang, zhongangqi, yingsshan}@tencent.com

## Abstract

Recent blind super-resolution (SR) methods typically consist of two branches, one for degradation prediction and the other for conditional restoration. However, our experiments show that a one-branch network can achieve comparable performance to the two-branch scheme. Then we wonder: how can one-branch networks automatically learn to distinguish degradations? To find the answer, we propose a new diagnostic tool – Filter Attribution method based on Integral Gradient (FAIG). Unlike previous integral gradient methods, our FAIG aims at finding the most *discriminative filters* instead of input pixels/features for degradation removal in blind SR networks. With the discovered filters, we further develop a simple yet effective method to predict the degradation of an input image. Based on FAIG, we show that, in one-branch blind SR networks, 1) we are able to find a very small number of (*1%*) discriminative filters for each specific degradation; 2) The weights, locations and connections of the discovered filters are all important to determine the specific network function. 3) The task of degradation prediction can be implicitly realized by these discriminative filters without explicit supervised learning. Our findings can not only help us better understand network behaviors inside one-branch blind SR networks, but also provide guidance on designing more efficient architectures and diagnosing networks for blind SR. The codes are available at https://github.com/TencentARC/FAIG.

## 1 Introduction

As a special category of image Super-Resolution (SR) [11, 6, 22], blind super-resolution [28, 3, 48] is an active research topic towards real-world restoration applications, and has attracted increasing attention. Blind SR aims at reconstructing a high-resolution image from its low-resolution counterpart which contains unknown and complex degradations (*e.g.*, blur, noise). Recent deep-learning-based blind SR methods typically consist of two branches: one for degradation prediction and the other for conditional restorations [14, 25, 39], as illustrated in Fig. 1. This design incorporates human knowledge about the degradation process [8, 23] and is consistent in classical blind SR methods [28, 11], thus implying a certain degree of interpretability.

With the powerful representation ability of deep learning, it is natural to wonder whether a unified one-branch network can effectively address the blind SR problem without the delicate two-branch

---

[*]Equal contributions. Liangbin Xie is an intern in ARC Lab, Tencent PCG.

35th Conference on Neural Information Processing Systems (NeurIPS 2021).

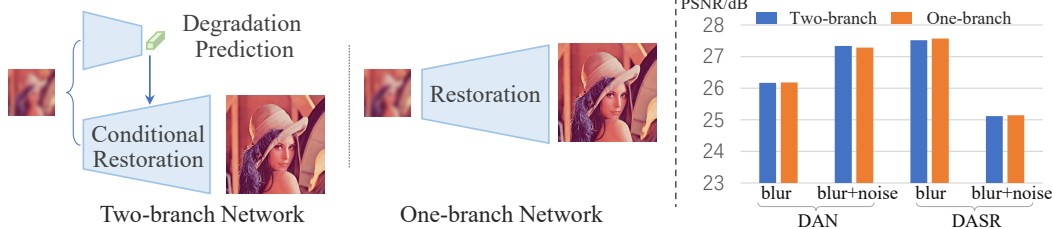

Figure 1: **Left**: Illustration for two-branch and one-branch blind SR networks. **Right**: A unified one-branch network could achieve comparable performance under similar computation budgets for state-of-the-art blind SR methods DAN [25] and DASR [39].

design from human prior knowledge. Thus, we conduct preliminary experiments on several state-of-the-art methods [25, 39] (more details in Sec. 3), and draw the observation that a unified one-branch network can achieve comparable results with similar computation budgets under proper training strategies, as shown in Fig. 1.

Despite its comparable performance, the one-branch network is more like a 'black-box', and we are curious about its connection with delicate two-branch designs with higher interpretability. Specifically, the phenomenon mentioned above motivates us to raise two key questions: **1)** Could one-branch networks automatically learn to distinguish degradations as what we specially design in two-branch methods? **2)** Are there any small sub-network (*i.e.*, a set of filters) existing for a specific degradation? For these open questions, we lack systematic investigations and analysis tools to gain further understandings.

In this paper, we make the first attempt to explore the underlying mechanism of blind SR networks. Our key findings are as follows: **1)** In one-branch blind SR networks, we are able to find a very small number of (*at least to 1%*) discriminative filters for each specific degradation (*e.g.*, blur, noise). When we mask these discovered filters for a specific degradation, the corresponding function is eliminated while functions for other degradations are maintained (as shown in Fig. 2, more details in Sec. 5.2.1). **2)** Except for the filter weights, the locations and connections of the discovered filters also play an essential role in the network function for a specific degradation (see Sec. 5.2.2). **3)** Based on these discovered filters, we could easily predict the degradation of input images without training in the supervision of degradation labels (see Sec. 5.3). In a word, a unified one-branch network is similar to a well-designed two-branch network in the working mechanism. The degradation prediction branch and several small sub-networks for various degradations could be automatically learned within the one-branch network, even it is trained without any supervision about degradation distinctions.

To analyze the mysterious blind SR networks and draw the above key findings, we propose a novel Filter Attribution method based on Integral Gradient (FAIG), aiming at finding *core filters* in a network that make the greatest contribution to the function of a specific degradation removal. Different from previous integral gradient methods [37, 36] that cumulate gradients along paths in the pixel/feature space, our FAIG utilizes paths in the *parameter space*, which has a clearer meaning in attributing network functional alterations to filter changes (*i.e.*, the changes of *filter parameters* rather than inputs result in the network function alteration). Built on FAIG, we further develop a simple yet effective method to predict the degradation of an input image (see Sec. 4.3).

To summarize, the contributions are three-fold. **1)** Our findings provide a better understanding of the mechanism under blind SR networks, especially, bringing insightful connections between popular two-branch methods and unified one-branch networks. **2)** The discovered discriminative filters for specific degradations allow us not only to perform degradation prediction, but also achieve a controllable adjustment of restoration strength without introducing extra parameters. **3)** Exploiting the interpretability of blind SR would bring great significance for future works in i) designing more efficient architectures; ii) diagnosing an SR network, such as determining the boundary of network restoration capacity and improving algorithm robustness.

## 2 Related Work

**The super-resolution** field has witnessed a variety of developments since the pioneer work of SRCNN [6, 7]. Different architectures are proposed, such as deeper networks [17, 19, 15], residual blocks [20, 22, 51, 43, 41], recurrent architectures [18, 38], and attention mechanism [50, 4, 24, 27].

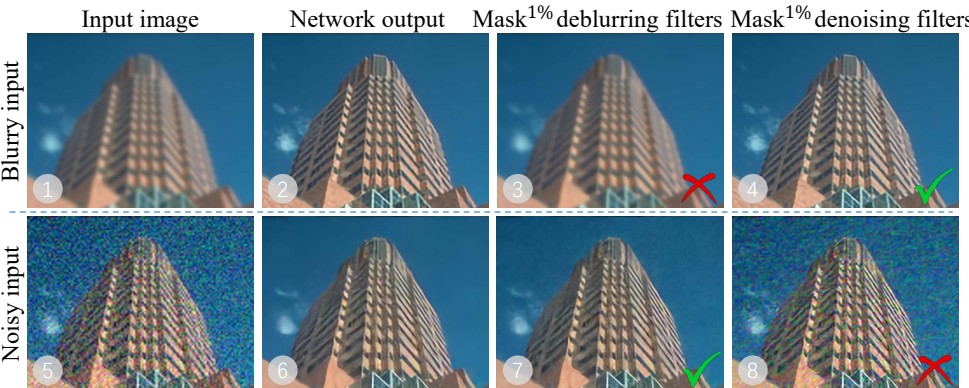

Figure 2: For a blurry input (①) and noisy input (⑤), the one-branch SRResNet [20] for blind SR could remove blur (②) and noise (⑥), respectively. When we mask the 1% deblurring filters (discovered by the proposed FAIG), the corresponding network function of deblurring is eliminated (③) while the function of denoising is maintained (⑦). Similar phenomenon happens (④ and ⑧) when we mask the 1% denoising filters in the same network. **Zoom in for best view**.

Most of these methods are unified one-branch networks and achieve excellent performance on SR with a fixed downsampling (*e.g.*, bicubic) kernel.

Recently, blind SR has attracted increasing attention to solve real-world restoration problems with complicated degradations. Classical blind SR methods typically incorporate human understanding of degradation process [8, 23], and are divided into two stages: degradation estimation (especially blur kernels) [28] and super-resolution based on the estimated degradation [11, 45]. Deep-learning-based methods inherit this design and develop two-branch blind SR networks. KernelGAN [3] leverages the internal distribution of image patches to estimate blur kernel. DASR [39] learns degradation representation and prediction with contrastive learning. SRMD [48] and [49, 46] incorporate these estimated degradations into CNN networks for restoration. IKC [14] and DAN [25] further propose to jointly perform degradation prediction and restoration in an iterative manner. Recent works also attempt to address blind SR within a one-branch networks [40, 47]. There are few related works investigating the relationship between two-branch and one-branch networks for blind SR. Moreover, the underlying mechanism behind blind SR is still mysterious.

**Network interpretation** methods for high-level visions can be divided into two categories: gradient-based methods [44, 33, 35, 34, 1, 53, 32] and perturbation-based approaches [9, 52, 30, 31]. The latter usually localizes the discriminative image regions by performing perturbation to the input or neurons [10]. They heavily depend on lots of samplings and resources to guarantee performance. The most relevant gradient-based method to ours is Integrated Gradients (IG) [37, 36], which accumulates gradients at all points along a path from the baseline input to target input. IntInf [21] and neuron conductance [5] further extend feature-important IG to neurons. Unlike these methods that cumulate gradients along paths in the pixel/feature space, our FAIG utilizes paths in the parameter space for attributing network functional alterations to filter changes. Recently, Gu and Dong [13] introduce an interpretation method with local attribution maps to SR networks, while we further try to uncover the underlying mechanism in blind SR networks.

**Interpolation in the network parameter space** has been used to explore the generalization and flat property of loss landscapes [12, 16]. It is also employed in image manipulation for continuous imagery effect transition [42]. In this work, we utilize the paths in parameter space (with interpolation) to find discriminative filters for specific degradations.

## 3 Preliminary

**Blind SR** aims to restore high-resolution images from low-resolution ones with unknown and complex degradations, *e.g.*, blur, noise and downsampling. The classical degradation model [8] is usually adopted to synthetic the low-resolution input. Generally, the ground-truth image $x^{gt}$ is first convolved with Gaussian blur kernel $k$. Then, a downsampling operation with scale factor $r$ is performed. Finally, the low-resolution $x$ is obtained by adding white Gaussian noise $n$:

$$x = (x^{gt} \circledast k) \downarrow_r + n. \tag{1}$$

Table 1: PSNR (on RGB channels) comparisons between two-branch and one-branch networks on blind SR.

| PSNR (dB) | DAN [25] | | DASR [39] | |
|---|---|---|---|---|
| | blur | blur+noise | blur | blur+noise |
| Official two-branch | 26.168±0.009 | 27.341±0.072 | 27.518±0.034 | 25.116±0.012 |
| SRResNet one-branch | 26.182±0.011 | 27.288±0.027 | 27.573±0.010 | 25.143±0.013 |

Recent blind SR methods including IKC [14], DAN [25] and DASR [39] typically adopt a two-branch design, as shown in Fig. 1. We then compare the two-branch design with commonly-used one-branch networks on two state-of-the-art methods: DAN and DASR. For a fair comparison, all experiments are conducted with their corresponding officially released codes, and the settings remain unchanged except the network structures. Specifically, SRResNet without batch normalization [20] is employed as the one-branch network. We adjust the number of residual blocks to match a similar computation budget (FLOPs and parameters) to two-branch methods.

We compare them on the blur and blur+noise settings (more specifications are in the supp. material). The PSNR comparisons are shown in Tab. 1 and Fig. 1. It is observed that a unified one-branch network can also achieve comparable results under similar computation budgets on both settings. This interesting observation motivates us to investigate the underlying mechanism of blind SR networks, especially one-branch networks. It is worth noting that we do not claim that one-branch networks are always better than two-branch designs. The two-branch methods that incorporate domain-specific knowledge may have an advantage in training stability and robustness. In this paper, this observation is our primal motivation for interpreting blind SR networks.

**Integrated Gradient** (IG) method [37, 36] is usually used in high-level tasks (*e.g.*, classification network). It attributes the most important input components (*e.g.*, pixels in input images) that affect the network predictions. Suppose we have a deep classification network $F : \mathbb{R}^d \mapsto [0, 1]$. Let $x \in \mathbb{R}^d$ be the input image and $\bar{x} \in \mathbb{R}^d$ be the baseline input. $\bar{x}$ is usually a black image or blurry images. The IG method accumulates the gradients at all points along a straight-line path in $\mathbb{R}^d$ from the baseline $\bar{x}$ to the input $x$. Formally, the integrated gradient along the $i^{th}$ dimension for an input $x$ and baseline $\bar{x}$ is defined as:

$$\mathtt{IG}_i(x) = (x_i - \bar{x}_i) \times \int_{\alpha=0}^1 \frac{\partial F(\bar{x} + \alpha \times (x - \bar{x}))}{\partial x_i} d\alpha, \tag{2}$$

where $x_i$ is its $i^{th}$ pixel value and $\frac{\partial F(x)}{\partial x_i}$ is the gradient of $F$ along the $i^{th}$ dimension at $x$. Each change in the input component adds up to the final prediction difference $\sum_i \mathtt{IG}_i(x) = F(x) - F(\bar{x})$.

Intuitively, the function $F$ varies from a less informative value (produced by the baseline $\bar{x}$) to its final informative value. The gradients of $F$ with respect to the image pixels explain each step of the variation in the value of $F$. The integration over the path gradients cumulates these small attributions and accounts for the difference of network outputs. In other words, the IG could interpret deep networks by finding out the most important input components (or features) that explain the network predictions.

## 4 Method

### 4.1 Filter Attribution Integrated Gradients

Our goal is to reveal the underlying mechanism of one-branch blind SR networks and answer the question: how can one-branch networks automatically learn to distinguish degradation. Specifically, we aim to find a small set of *core filters* [2] that could explain the network functions of a specific degradation removal.

Different from the IG method that attributes the network output changes to the inputs or feature, we propose Filter Attribution Integrated Gradients (FAIG) to attribute network functional alterations to filter changes. Inspired by IG, our key idea is that: the baseline network is a pure SR network that cannot remove any degradations, while the target network is a re-trained network that can deal with complex degradations. Given the same input, the changes of the network output can be attributed to the changes of network parameters (*i.e.*, filters).

Let $F(\theta, x)$ be the blind SR network with parameters $\theta$ and input $x$. We quantify the network function of degradation removal by:

$$\mathcal{L}(\theta, x) = \|F(\theta, x) - x^{gt}\|_2^2, \tag{3}$$

---

[2] In this paper, we define each $K \times K$ weight in a convolutional layer as a filter, where $K$ is the kernel size.

where $x^{gt}$ is the ground-truth of $x$. $\mathcal{L}(\theta, x)$ measures the distance between the network output and the ground-truth. Lower distances indicate a stronger degradation removal function.

Let $\gamma(\alpha), \alpha \in [0, 1]$ be a continuous path between the baseline model and the target model, satisfying $\gamma(1) = \bar{\theta}, \gamma(0) = \theta$. For an input image $x$, we then attribute the changes of network functions to the changes of filters with path integrated gradient as follows:

$$\mathcal{L}(\bar{\theta}, x) - \mathcal{L}(\theta, x) = \mathcal{L}(\gamma(1), x) - \mathcal{L}(\gamma(0), x) \tag{4}$$
$$= \sum_i \int_{\alpha=0}^{1} \frac{\partial \mathcal{L}(\gamma(\alpha), x)}{\partial \gamma(\alpha)_i} \times \frac{\gamma(\alpha)_i}{\partial \alpha} d\alpha,$$

Then, the $i^{th}$ dimension (*i.e.*, different network parameters) of the FAIG could be defined as:

$$\texttt{FAIG}_i(\theta, x) = \int_{\alpha=0}^{1} \frac{\partial \mathcal{L}(\gamma(\alpha), x)}{\partial \gamma(\alpha)_i} \times \frac{\gamma(\alpha)_i}{\partial \alpha} d\alpha \tag{5}$$

As described in [29, 13], the choice of baselines and path functions affect the final attribution results. We now present the choice for the baseline model and integrated path.

**Baseline model.** A good baseline model should have the following properties: 1) The baseline should represent the 'absence' of the desired function, *i.e.*, not having the corresponding degradation removal function in our case. 2) The output of baseline models should also be an image with the same content as the input, otherwise, the discovered filters may capture degradation-irrelevant changes. 3) The baseline model is better to locate in a smaller neighborhood around the target model, so that no drastic changes will happen and fewer core filters will be found.

To satisfy the above properties, we propose a fine-tuning strategy to construct such a pair of models. Specifically, we first train a common SR model for the bilinear downsampling kernel as the baseline model $F(\bar{\theta})$, which could not resolve degradations such as blur and noise. Then, we fine-tune it to obtain $F(\theta)$ for downsampling, blur and noise together, as the target model.

**Integrated path.** Instead of cumulating gradients along paths in the pixel/feature space, our FAIG utilizes paths in the parameter space. A direct yet effective path is a straight-line path between $F(\bar{\theta})$ and $F(\theta)$. Formally, the path could be represented by $\gamma(\alpha) = \alpha\bar{\theta} + (1 - \alpha)\theta$, where $\gamma(1) = \bar{\theta}$, $\gamma(0) = \theta$.

Thus, the Eq. 5 can be re-written as follows. In practice, the integral can be approximated via discrete points uniformly sampled along the path.

$$\texttt{FAIG}_i(\theta, x) = \left[\bar{\theta} - \theta\right]_i \int_{\alpha=0}^{1} \left[ \frac{\partial \mathcal{L}(\gamma(\alpha), x)}{\partial \gamma(\alpha)} \bigg|_{\gamma(\alpha)=\theta+\alpha\times(\bar{\theta}-\theta)} d\alpha \right]_i \tag{6}$$
$$\approx \frac{1}{N} \left[\bar{\theta} - \theta\right]_i \sum_{k=0}^{N-1} \left[ \frac{\partial \mathcal{L}(\gamma(\alpha), x)}{\partial \gamma(\alpha)} \bigg|_{\gamma(\alpha)=\theta+\alpha\times(\bar{\theta}-\theta),\alpha=k/N} \right]_i,$$

where $N$ is the total steps in the approximation of the integral, and we empirically set it to 100.

**Discussions.** *Comparing with directly finding filters with the largest changes of absolute values.* Given the baseline model and the target model, one may propose to directly calculate the absolute value changes of parameters to determine the most important filters for the network functional alteration, *i.e.*, $|\theta - \bar{\theta}|_i$. However, such a method does not consider the influence of network outputs, thus the filter changes may result in other dimensions of transition, instead of the degradation removal function. Moreover, it cannot discover different filter sets for various degradations inside one model.

*Difference from IntInf [21] and neuron conductance [5].* IntInf and neuron conductance are integral gradient methods that measure feature importance with respect to inputs and filters together. Yet, they still alter the input from a baseline $\bar{x}$ to the input at hand $x$, and then calculate the gradients with respect to filters along this path in the input space. Therefore, their gradient calculation cannot directly attribute function alteration to filters. We show the superior performance of our method in Sec. 5.2.1.

## 4.2 Finding Discriminative Filters for a Specific Degradation

The above `FAIG` is capable of finding important filters for one degradation, but it still has two limitations: 1) the discovered filters do not guarantee to be *only* responsible for this degradation. Specifically, the filters found for the deblurring function may also have functions for other degradations. 2) Eq. 6 is calculated for a single input image, while we want to find discriminative filters that are only correlated with degradations but not image contents.

Therefore, we further propose two improvements. 1) The intuitive idea of finding filters *only* responsible for one degradation is to suppress gradients to other degradations while retaining the interested gradients. Thus, for the $i^{th}$ parameter, we calculate the gradient difference between a specific degradation of interest $\mathcal{D}$ and other degradations $\sim \mathcal{D}$. 2) We average all the gradient difference in a whole dataset to eliminate the impact of image contents. Formally, we have:

$$\text{FAIG}_i^{\mathcal{D}}(\theta) = \frac{1}{|\mathcal{X}|} (\underbrace{\sum_{x \in \mathcal{X}} |\text{FAIG}_i(\theta, x^{\mathcal{D}})|}_{\text{attribution for degradtion} \mathcal{D}} - \underbrace{\sum_{x \in \mathcal{X}} |\text{FAIG}_i(\theta, x^{\sim \mathcal{D}})|}_{\text{attribution for other degradtions}} ), \qquad (7)$$

where $\mathcal{D}$ denotes a specific degradation while $\sim \mathcal{D}$ denotes the other degradations; $\mathcal{X}$ is the dataset that we are used for calculating FAIG. We then sort the obtained $\text{FAIG}_i^{\mathcal{D}}(\theta)$ in descending order, and get the top filters by a certain percentage. Those discovered filters are the discriminative filters for a specific degradation $\mathcal{D}$.

## 4.3 Degradation Prediction

Based on the discovered filters, we further develop a simple yet effective method to predict the degradation of an input image. Specifically, let $\{\text{filter}^{\mathcal{D}}\}$ be the set of discovered filters for degradation $\mathcal{D}$ (found by Eq.7), and let $\{\text{filter}^x\}$ be the set of discovered filters for input image $x$ (found by Eq.6). Then, we calculate the overlap score (OS) to measure the intersection of the two sets of filters:

$$\text{OS}(x, \mathcal{D}) = \frac{|\{\text{filter}^{\mathcal{D}}\} \cap \{\text{filter}^x\}|}{|\{\text{filter}^x\}|}. \qquad (8)$$

Higher overlap score $\text{OS}(x, \mathcal{D})$ represents that the degradation of input image $x$ is more similar to the degradation $\mathcal{D}$. Thus, we are able to predict the degradation of input images by comparing the $\text{OS}(x, \mathcal{D})$ with a threshold $\mathsf{T}^{\mathcal{D}}$ for different degradations. The threshold could be determined by a validation dataset.

# 5 Experiments

## 5.1 Implementation Details

**Network architecture.** Our analyses are conducted on two representative network architectures: 1) SRCNN-style [7] with nine convolutional layers except that the upsampling is at the end of the network; 2) SRResNet [20] without batch normalization layers. Our FAIG method could perform well on both of these networks. Networks with more complicated components such as attention modules [50, 4, 24] are left as future work.

**Degradations.** As the first work that interprets blind SR networks, we mainly consider the bilinear downsampling with a scale factor of 2, Gaussian blur with a sigma of 2, and Gaussian noise with a level of 0.1. We use Eq. 1 to generate degraded images. Blur and noise are applied independently, and they are applied with a probability of 0.5.

**Datasets.** We train and fine-tune our models on the DIV2K training dataset [2], which consists of 800 high-resolution images. The analyses are conducted on Set14 [45], BSD100 [26], and DIV2K validation set (100 images) [2]. All these datasets are commonly used in SR and licensed for research purposes.

**Training details.** We first train a network with common bilinear downsampling to obtain the baseline model. Then, we fine-tune it to obtain the target model that could solve bilinear downsampling, blur and noise together. The initial learning rate for fine-tuning is set to $5 \times 10^{-5}$. The optimization is

Table 2: We compare the performance drop with other methods. For blurry (noisy) inputs, we mask the corresponding deblurring (denoising) filters. Larger values indicates a large performance drop. Test on Set14.

| $(10^{-3})$ | mask 1% discovered filters | | | | mask 5% discovered filters | | | |
|---|---|---|---|---|---|---|---|---|
| Input | FAIG (ours) | IG | $\|\theta - \bar{\theta}\|$ | Random | FAIG (ours) | IG | $\|\theta - \bar{\theta}\|$ | Random |
| Blurry image | **6.68**±0.63 | 4.31±1.54 | 0.18±0.13 | 0.07±0.01 | **7.53**±0.24 | 6.41±0.88 | 2.16±0.61 | 0.55±0.32 |
| Noisy image | **6.62**±0.54 | 4.22±0.44 | 0.49±0.10 | 0.04±0.01 | **16.28**±3.84 | 8.01±1.04 | 3.25±1.85 | 0.19±0.05 |

performed with Adam optimizer for $100K$ iterations. Training batch size is set to 16 and training patch size of ground-truth images is set to $64 \times 64$. All the training and analyses are performed with PyTorch on NVIDIA V100 GPUs in an internal cluster.

We also run each experiment and analysis for three different random seeds and report error bars in our results.

## 5.2 Evaluations and Analyses

As the filters in SR networks closely relate to each other, simply deleting or inserting filters will introduce severe artifacts to outputs. Considering that, in FAIG, there are two strongly connected models: a target model $F(\theta)$ and a baseline model $F(\bar{\theta})$, that locate closely in the parameter space. The former could effectively remove degradations while the latter cannot. Such nice properties could be utilized to inspect the function alteration of networks. Based on this, we propose two ways to verify the effectiveness of finding discriminative filters for specific degradations by FAIG (see Sec. 5.2.1 and Sec. 5.2.2). We further analyze the distribution of discovered filters in Sec. 5.2.3.

### 5.2.1 Mask Discovered Filters

We measure the importance of discovered filters by replacing them with the filters in the baseline model (at the same locations). After substitution, we could obtain a new model. Then, given different degraded images (*e.g.*, blurry or noisy images) as inputs, we inspect the model outputs. If those discovered filters play an important role in removing a specific degradation, the new substituted model will lose such a function and could not remove the corresponding degradation. We name this method as 'masking' the functionality of discovered filters.

Thus, we could quantify the discovered filters' contribution to the network function by measuring *output differences* (*e.g.*, MSE) of target model $F(\theta)$ and the substituted model. In order to reduce the effects of brightness and contrast changes, we calculate output differences on image gradients of their gray counterparts. A large difference represents a large performance drop caused by the replacement of discovered filters, implying that those filters have strong effects on a specific network function.

The results of masking discovered filters with different proportions are shown in Fig. 3. We can draw the following observations. **1)** When we mask FAIG-discovered filters for deblurring (even a very small portion), the performance for deblurring drops drastically (①) while the function of denoising is maintained (②) at small portions and then has a slow performance drop for most proportions, and finally has a quick decline for very large proportions. We are more interested in the small proportion with discriminative filters, as they are more informative for interpretation. It is also similar to mask filters for denoising (③ and ④). **2)** If we randomly select a portion of filters to mask, the network shows similar performance drops for all cases, implying that the randomly selected filters are non-discriminative. **3)** For the small proportion (*e.g.*, 1%), the randomly selected filters could not have a large impact on network function alteration, obviously shown in the illustrative images. Note that Fig. 3 shows the results of SRResNet, and SRCNN-style networks have similar phenomenons (see supplementary materials).

We also compare our FAIG with other methods: modified IG (*i.e.*, IntInf [21]), absolute values of filter changes ($\|\theta - \bar{\theta}\|$), and random selection. We measure the MSE error of image gradients as above. As shown in Tab. 2, our FAIG can discover filters that result in larger performance drop, *i.e.*, discover more important filters for corresponding degradations.

### 5.2.2 Re-train only Discovered Filters

With the method of masking filters, we could conclude that the *weights* of discovered filters are essential for network functions. We are also curious whether *the locations and connections* of the discovered filters are important. Thus, we further examine the FAIG-discovered filters in another direction, *i.e.*, from the baseline model, we re-train *only* the discovered filters for blind SR. Specifically,

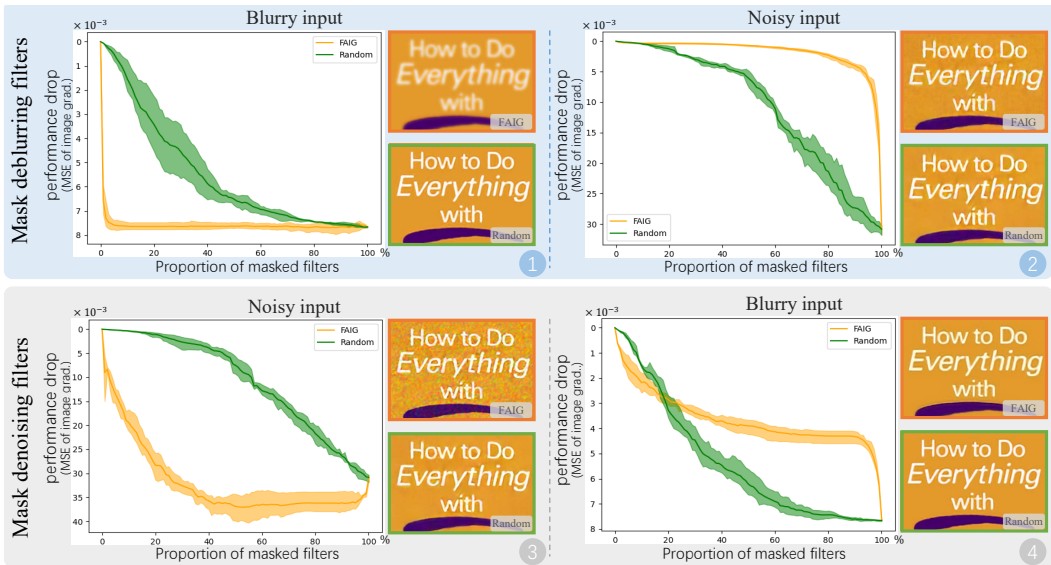

Figure 3: Results of masking discovered filters (by FAIG and random selection) with different proportions (on SRResNet). The shown images are produced by 1% mask. **Zoom in for best view**.

Table 3: Results of re-training baseline models with 1% filters for deblurring and denoising. Test on Set14.

| PSNR(dB) | | Re-train 1% filters for deblurring | | | | Re-train 1% filters for denoising | | | |
|---|---|---|---|---|---|---|---|---|---|
| Input | Upper bound | FAIG | IG | $\lvert \theta - \bar{\theta} \rvert$ | Random | FAIG | IG | $\lvert \theta - \bar{\theta} \rvert$ | Random |
| Blurry | 29.203 ($\pm$0.021) | **27.889** ($\pm$0.207) | 26.389 ($\pm$0.274) | 26.444 ($\pm$0.097) | 26.691 ($\pm$0.092) | 27.642 ($\pm$0.007) | 26.534 ($\pm$0.125) | 26.444 ($\pm$0.096) | 26.668 ($\pm$0.126) |
| Noisy | 26.712 ($\pm$0.008) | 25.268 ($\pm$0.035) | 25.211 ($\pm$0.005) | 25.288 ($\pm$0.044) | 25.239 ($\pm$0.034) | **25.743** ($\pm$0.033) | 25.141 ($\pm$0.116) | 25.275 ($\pm$0.035) | 25.204 ($\pm$0.016) |

1) we first find the discriminative filters for a degradation $\mathcal{D}$ on the target model, and record the locations of those filters. 2) Re-train the baseline model on the desired degradation. Note that we only re-train the corresponding filters *with the same locations* in the baseline model, while keeping other filters unchanged.

We compare the performance (measured by PSNR) of the re-trained models. These models are different in the ways of finding such discriminative filters - 1) FAIG, 2) modified IG (*i.e.*, IntInf [21]), 3) absolute values of filter changes, *i.e.*, $\lvert \theta - \bar{\theta} \rvert$, 4) random selection. We also include the upper bound that re-train all parameters of the baseline model.

As shown in Tab. 3, when we only re-train deblurring filters for the deblurring task, our FAIG has a large performance gain (than random selection). Yet, the performance of FAIG is close to that of random selection when we only re-train denoising filters for the deblurring task. This implies that the locations and connections of discovered filters also have discriminative characteristics for specific degradations. It is also interesting to observe that re-training filters found by other methods leads to inferior (or similar) performance than random selection. We conjecture that the locations and connections discovered by those methods may hinder the re-training process, which requires further investigation in future work.

The above observations has shown that the weights, locations and connections of the discovered filters are all important to determine the network function for a specific degradation.

### 5.2.3 Distribution of Discovered Filters in a Network

We then delve into the discovered filters for deeper observations. We visualize the discovered 1% filters for different network functions in Fig. 4. It is observed that the filters for deblurring and denoising have very different distributions inside the network. The deblurring filters are more located in the back part of the network while denoising filters locate more uniformly. We conjecture that the deblurring operation requires a larger receptive field for an equivalent 'deconvolution' kernel. Moreover, their complementary distribution also implies the division of labor in a network.

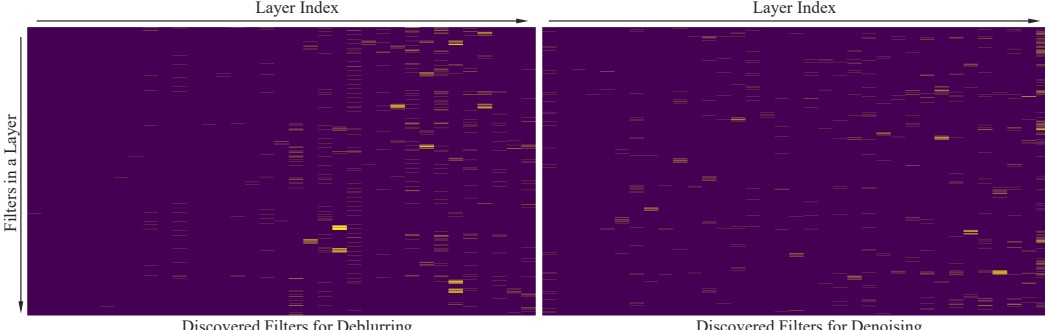

Figure 4: Visualization of the discovered 1% filters (yellow lines) in a blind SRResNet network for deblurring and denoising. (Yellow lines are thickened for better visualization while not affecting the distribution). It is observed that the sub-networks for deblurring and denoising have a very different distribution inside the network, implying their division of labor. **Zoom in for best view**.

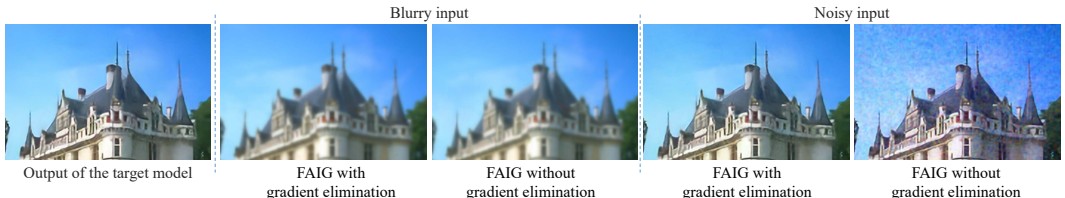

Figure 5: Importance of gradient elimination for other degradations. We mask 1% deblurring filters in this comparison. **Zoom in for best view**.

## 5.3  Degradation Classification

We adopt Set14 to discover filters for a specific degradation $\mathcal{D}$. The thresholds are then calculated on the BSD100 dataset. Finally, the degradation prediction is performed on the DIV2K dataset. We set $\mathsf{T}^{noise}$ and $\mathsf{T}^{blur}$ to 0.6 and 0.5, and the prediction accuracy can reach $98\%$ and $96\%$, respectively (more results are in the supp. material). Therefore, the task of degradation prediction can be implicitly realized by these discriminative filters without explicit supervised learning.

## 5.4  Importance of Gradient Elimination for Other Degradations

In Sec. 4.2, we introduce an improvement of gradient elimination for other degradations. Such a design is important to find discriminative filters for a specific degradation. The comparison is shown in Fig. 5. We mask 1% deblurring filters in this comparison. For the blurry input, both models can produce expected results (*i.e.*, lose the deblurring function). While for the noisy input, FAIG without gradient elimination for other degradations could also lose the denoising function, indicating that the discovered filters are not discriminative. In contrast, our FAIG with gradient elimination for other degradations could also maintain its ability to denoising.

## 5.5  Limitations

Our work has several limitations. 1) The setting of considered degradations in our analyses is a simple one in blind SR. It only considers one discrete level for each degradation. Future work should take more complex and practical degradations into consideration. 2) Our analyses are conducted in SRCNN-style and SRResNet. However, recent progress in network architectures such as attention modules is necessary to analyze in blind SR. 3) There is still a gap to apply our findings to design more efficient networks and diagnose networks for blind SR. We believe that our investigation is inspiring and the above limitations should be addressed in future works.

# 6  Conclusion

In this paper, we make the first attempt to investigate the mechanism underlying the unified one-branch blind SR network. We propose a new diagnostic tool – Filter Attribution method based on Integral Gradient (FAIG) that utilizes paths in the parameter space in attributing network functional alterations to filter changes. Based on the proposed FAIG, we are able to find a very small number of (at least to 1%) discriminative filters for specific degradations in blind SR. We also show that

the weights, locations and connections of the discovered filters are all important to determine the specific network function. Moreover, the task of degradation prediction can be implicitly realized by these discriminative filters without explicit supervised learning. Our findings can help us better understand network behaviors inside one-branch blind SR networks. We believe that exploiting the interpretability of blind SR would bring great significance for future works in designing more efficient architectures and diagnosing an SR network, such as determining the boundary of network restoration capacity and improving algorithm robustness.

## 7 Acknowledgements

This work is partially supported by National Natural Science Foundation of China (61906184), the Science and Technology Service Network Initiative of Chinese Academy of Sciences (KFJ-STS-QYZX-092), the Shanghai Committee of Science and Technology, China (Grant No. 21DZ1100100).

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
