# Supplementary File for
# Finding Discriminative Filters for Specific Degradations in Blind Super-Resolution

**Liangbin Xie**[*1,2,3]     **Xintao Wang**[*3]     **Chao Dong**[1,4]     **Zhongang Qi**[3]     **Ying Shan**[3]

[1]Shenzhen Key Lab of Computer Vision and Pattern Recognition,
Shenzhen Institute of Advanced Technology, Chinese Academy of Sciences
[2]University of Chinese Academy of Sciences
[3]ARC Lab, Tencent PCG
[4]Shanghai AI Laboratory, Shanghai, China
{lb.xie, chao.dong}@siat.ac.cn
{xintaowang, zhongangqi, yingsshan}@tencent.com

## Abstract

In this supplementary file, we first show the details about the comparison between one-branch and two-branch blind SR networks in Section 1. More experiments and analyses of masking discovered filters in SRCNN-style and SRResNet networks are provided in Section 2 and Section 3, respectively. The impact of choosing different baseline models in FAIG is discussed in Section 4. We provide more results of degradation prediction in Section 5. Finally, we present a controllable adjustment of restoration strength with the discovered degradation-specific filters in Section 6. Furthermore, we include the core codes of FAIG in this supplementary file. The codes are available at https://github.com/TencentARC/FAIG.

## 1 Details about the comparison between two-branch and one-branch blind SR networks

DAN [3] and DASR [5] are two state-of-the-art methods in blind SR. The FLOPs and parameters of these two models are listed in Tab. 1. For a fair comparison, we adjust the number of residual blocks in SRResNet to match a similar computation budget to these methods. There are two alternatives for measuring computation budgets - FLOPs and the total number of parameters. We can adjust the number of residual blocks to have similar FLOPs or have a similar number of parameters. And we choose the smaller network between these two alternatives for a more convincing conclusion. Specifically, the one-branch SRResNet for DAN keeps nearly the same parameters as that of DAN, while the one-branch SRResNet for DASR keeps nearly the same FLOPs as that of DASR.

We use the officially released codes for DAN [2] and DASR [3]. The experiment settings remain unchanged except the network structures (shown in Tab. 1). Note that DASR first trains the degradation representation with MOCO for 100 epochs, and then trains the whole network (degradation representation and conditional restoration) for 500 epochs. For a fair comparison, we train the corresponding one-branch SRResNet for 500 epochs.

We compare their performance on the blur and blur+noise settings. We adopt the same evaluation settings as those in their papers. Specifically, for the blur setting, we sample 8 kernels from the range

---

[*]Equal contributions. Liangbin Xie is an intern in ARC Lab, Tencent PCG.
[2]https://github.com/greatlog/DAN
[3]https://github.com/LongguangWang/DASR

[1.8, 3.2] for DAN, while choosing blur kernel width $\sigma = 1.2$ for DASR. The blur+noise setting adopted in our experiments is the same as that in DAN and DASR.

Our runs with the officially released training codes can reach the performance as reported in their paper. In the blur setting on $\times 4$ blind SR tested on the BSD100 dataset, 1) the DAN result of our runs in Y channel is **27.52dB**, which is very close to the reported performance in the DAN paper (**27.51dB** in Table 1.). 2) the DASR result of our runs is **27.52dB**, which is the same as the reported performance in the DASR paper (**27.52dB** in Table 2.)

The comparison results are shown in Tab. 2. The official two-branch and one-branch SRResNet networks are trained three times with different random seeds. For both blur and blur+noise settings, we can observe that the one-branch SRResNet network achieves comparable results with their corresponding two-branch networks under similar computation budgets.

Table 1: FLOPs, parameters and training details of DAN, DASR and their corresponding one-branch SRResNet networks with similar computation budgets.

| Methods | FLOPs | Parameters | Scale | Batch size | Initial learning rate | Total iterations (epochs) |
|---|---|---|---|---|---|---|
| DAN official | 275.36G | 4.33M | $\times 4$ | 64 | $2 \times 10^{-4}$ | $4 \times 10^5$ iterations |
| SRResNet one-branch | 77.03G | 4.32M | $\times 4$ | 64 | $2 \times 10^{-4}$ | $4 \times 10^5$ iterations |
| DASR official | 48.27G | 7.25M | $\times 4$ | 64 | $1 \times 10^{-3}$ | 600 epochs |
| SRResNet one-branch | 48.32G | 2.33M | $\times 4$ | 64 | $1 \times 10^{-3}$ | 500 epochs |

Table 2: PSNR (on RGB channels) comparisons between two-branch and one-branch networks on blind SR.

| PSNR (dB) | DAN [3] | | DASR [5] | |
|---|---|---|---|---|
| | blur | blur+noise | blur | blur+noise |
| Official two-branch | 26.168±0.009 | 27.341±0.072 | 27.518±0.034 | 25.116±0.012 |
| SRResNet one-branch | 26.182±0.011 | 27.288±0.027 | 27.573±0.010 | 25.143±0.013 |

## 2  Mask discovered filters in SRCNN-style network

We present the result of masking discovered filers in the SRCNN-style network in this section. SRCNN-style [1] network has nine convolutional layers and the upsampling is operated at the end of the network. Fig. 1 shows the results of masking discovered filters in SRCNN-style with different properties. We can also observe the same observations as those of SRResNet (Fig. 3 in the main paper): **1)** When we mask FAIG-discovered filters for deblurring (even a very small portion), the performance for deblurring drops drastically (①) while the function of denoising is maintained (②) at small portions and then has a slow decline for large proportions. It is also similar to mask filters for denoising (③ and ④). **2)** If we randomly select a portion of filters to mask, the network shows similar performance drops, implying that the randomly selected filters are non-discriminative. **3)** For the small proportion (*e.g.*, 1%), the randomly selected filters could not have a large impact on network function alteration, obviously shown in the illustrative images.

Therefore, the discriminative filters exist in both the representative networks: plain SRCNN-style networks and SRResNet networks. Our FAIG can be applied to analyze networks with different structures.

**Comparison with other methods.** We compare our FAIG with other methods: modified IG (*i.e.*, IntInf [2]), absolute values of filter changes ($|\theta - \bar{\theta}|$), and random selection. We measure the MSE error of image gradients as described in Sec. 5.2.1 (in the main paper). A large error/difference represents a large performance drop caused by the replacement of discovered filters.

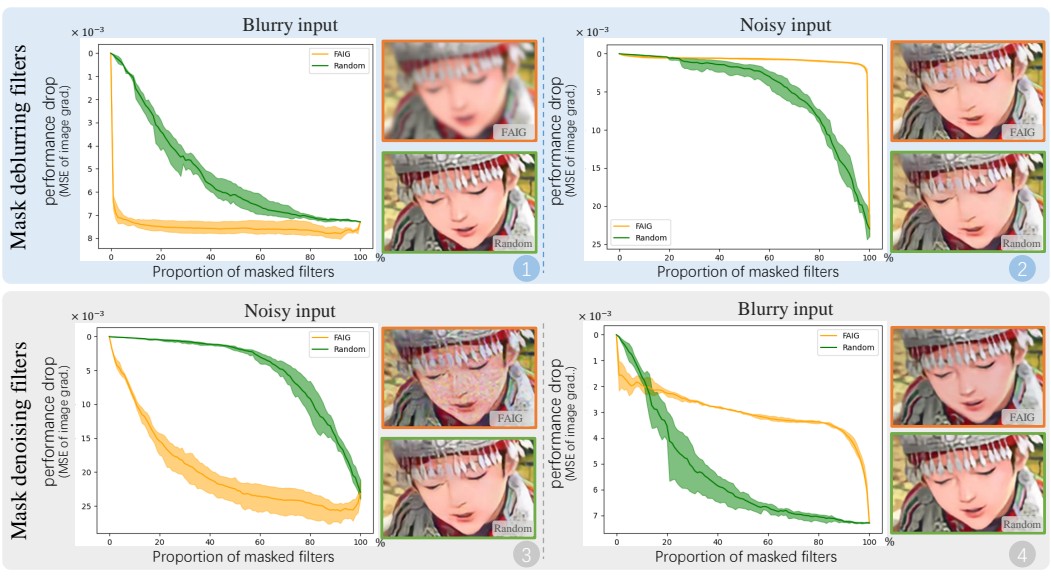

Figure 1: Results of masking discovered filters (by FAIG and random selection) with different proportions (on **SRCNN-style** network. The shown images are produced by 1% mask). **Zoom in for best view**.

The comparisons on Set14 [6] and BSD100 [4] are represented in Tab. 3 and Tab. 4, respectively. In both two datasets, our FAIG is able to discover filters that result in larger performance drop, *i.e.*, discover more important filters for corresponding degradations.

Table 3: We compare the performance drop with other methods. For blurry (noisy) inputs, we mask the corresponding debluring (denoising) filters. Larger values indicates a large performance drop, indicating discovering more important/discriminative filters for corresponding degradations. Test on **SRCNN-style** networks and **Set14**.

| $(10^{-3})$ | mask 1% discovered filters | | | | mask 5% discovered filters | | | |
|---|---|---|---|---|---|---|---|---|
| Input | FAIG (ours) | IG | $\|\theta - \bar{\theta}\|$ | Random | FAIG (ours) | IG | $\|\theta - \bar{\theta}\|$ | Random |
| Blurry image | **6.42**±0.30 | 0.86±0.41 | 0.91±0.70 | 0.07±0.03 | **7.18**±0.18 | 3.05±1.02 | 2.81±0.33 | 0.50±0.02 |
| Noisy image | **1.78**±0.05 | 0.45±0.32 | 0.63±0.31 | 0.04±0.02 | **5.35**±0.52 | 1.14±1.21 | 2.08±0.13 | 0.11±0.01 |

Table 4: We compare the performance drop with other methods. For blurry (noisy) inputs, we mask the corresponding debluring (denoising) filters. Larger values indicates a large performance drop, indicating discovering more important/discriminative filters for corresponding degradations. Test on **SRCNN-style** networks and **BSD100**.

| $(10^{-3})$ | mask 1% discovered filters | | | | mask 5% discovered filters | | | |
|---|---|---|---|---|---|---|---|---|
| Input | FAIG (ours) | IG | $\|\theta - \bar{\theta}\|$ | Random | FAIG (ours) | IG | $\|\theta - \bar{\theta}\|$ | Random |
| Blurry image | **5.02**±0.36 | 0.74±0.42 | 0.63±0.65 | 0.05±0.02 | **5.74**±0.29 | 2.53±1.02 | 2.18±0.25 | 0.45±0.07 |
| Noisy image | **1.40**±0.37 | 0.28±0.11 | 0.63±0.35 | 0.02±0.00 | **4.86**±0.90 | 0.93±1.05 | 2.01±0.12 | 0.11±0.01 |

**More qualitative results** of masking discovered filters with three different proportions (*i.e.*, 1%, 5%, 10%) are shown in Fig. 2, Fig. 3 (on Set14), and Fig. 4, Fig. 5 (on BSD100).

When we mask the deblurring filters, the corresponding network function of deblurring is eliminated (②) while the function of denoising is maintained (③). Similarly, when we mask the denoising filters, the corresponding network function of denoising is eliminated (④) while the function of deblurring is maintained (⑤). Moreover, the performance drop is larger when more discriminative filters are masked. In a word, benefiting from our FAIG, the discovered filters maintain the discriminative property for different degradations.

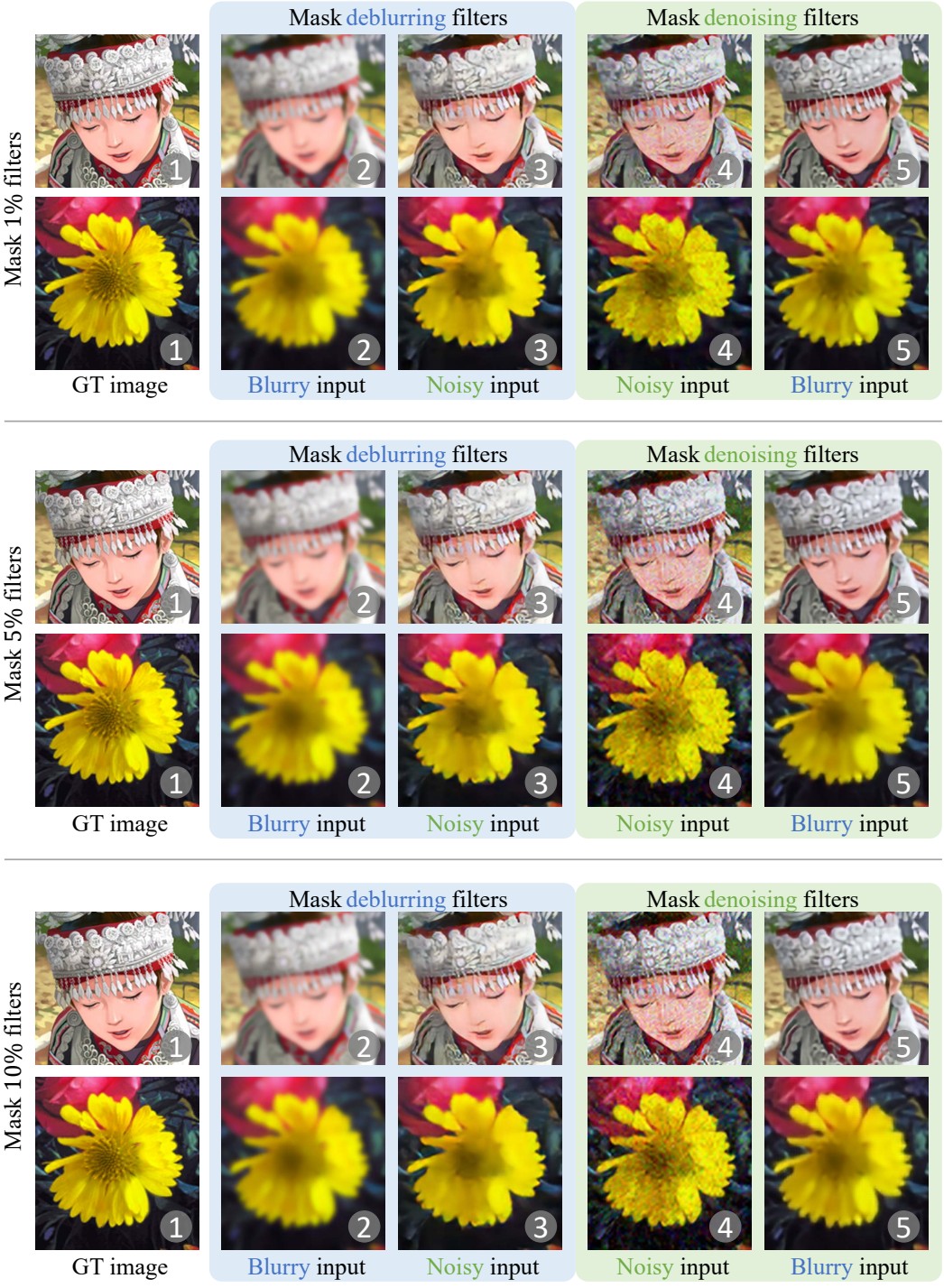

Figure 2: Qualitative results of masking different proportion (from top to bottom: 1%, 5%, and 10%) of discovered filters (by FAIG) in **SRCNN-style** network. Test on **Set14**. When we mask the deblurring filters, the corresponding network function of deblurring is eliminated (②) while the function of denoising is maintained (③). Similarly, when we mask the denoising filters, the corresponding network function of denoising is eliminated (④) while the function of deblurring is maintained (⑤). Moreover, performance drop is larger when more discriminative filters are masked. **Zoom in for best view**.

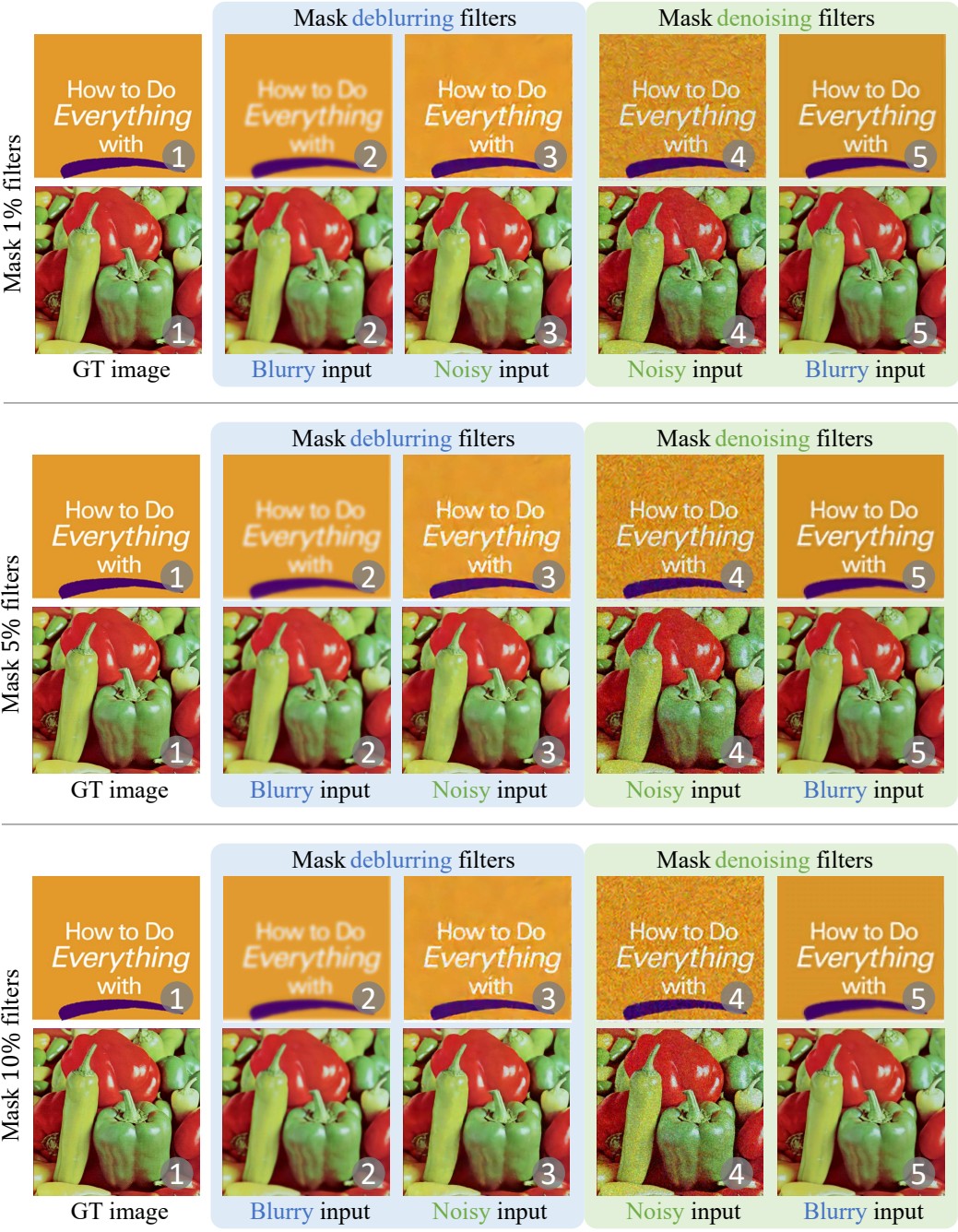

Figure 3: Qualitative results of masking different proportion (from top to bottom: 1%, 5%, and 10%) of discovered filters (by FAIG) in **SRCNN-style** network. Test on **Set14**. When we mask the deblurring filters, the corresponding network function of deblurring is eliminated (②) while the function of denoising is maintained (③). Similarly, when we mask the denoising filters, the corresponding network function of denoising is eliminated (④) while the function of deblurring is maintained (⑤). Moreover, the performance drop is larger when more discriminative filters are masked. **Zoom in for best view**.

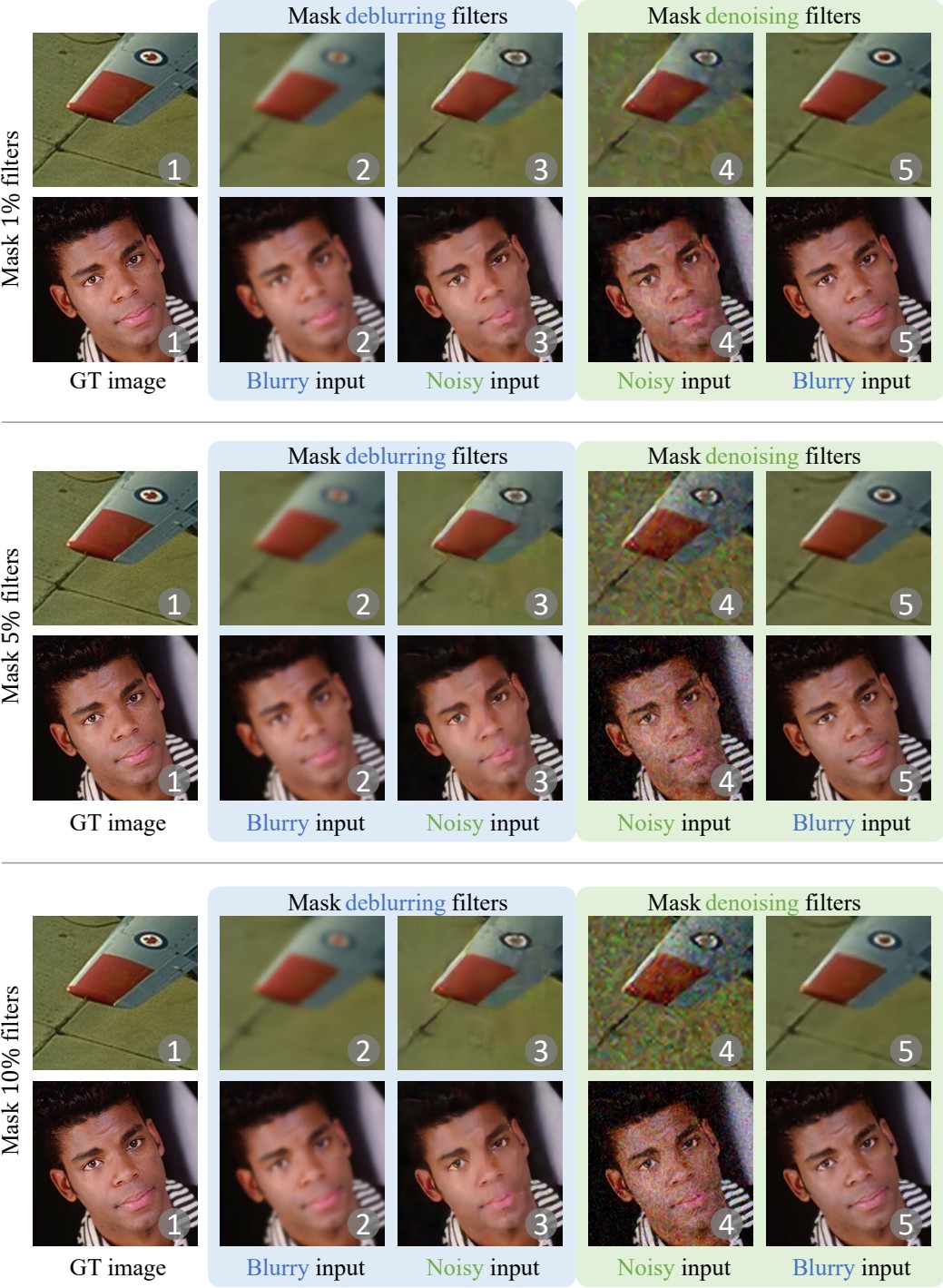

Figure 4: Qualitative results of masking different proportion (from top to bottom: 1%, 5%, and 10%) of discovered filters (by FAIG) in **SRCNN-style** network. Test on **BSD100**. When we mask the deblurring filters, the corresponding network function of deblurring is eliminated (②) while the function of denoising is maintained (③). Similarly, when we mask the denoising filters, the corresponding network function of denoising is eliminated (④) while the function of deblurring is maintained (⑤). Moreover, the performance drop is larger when more discriminative filters are masked. **Zoom in for best view**.

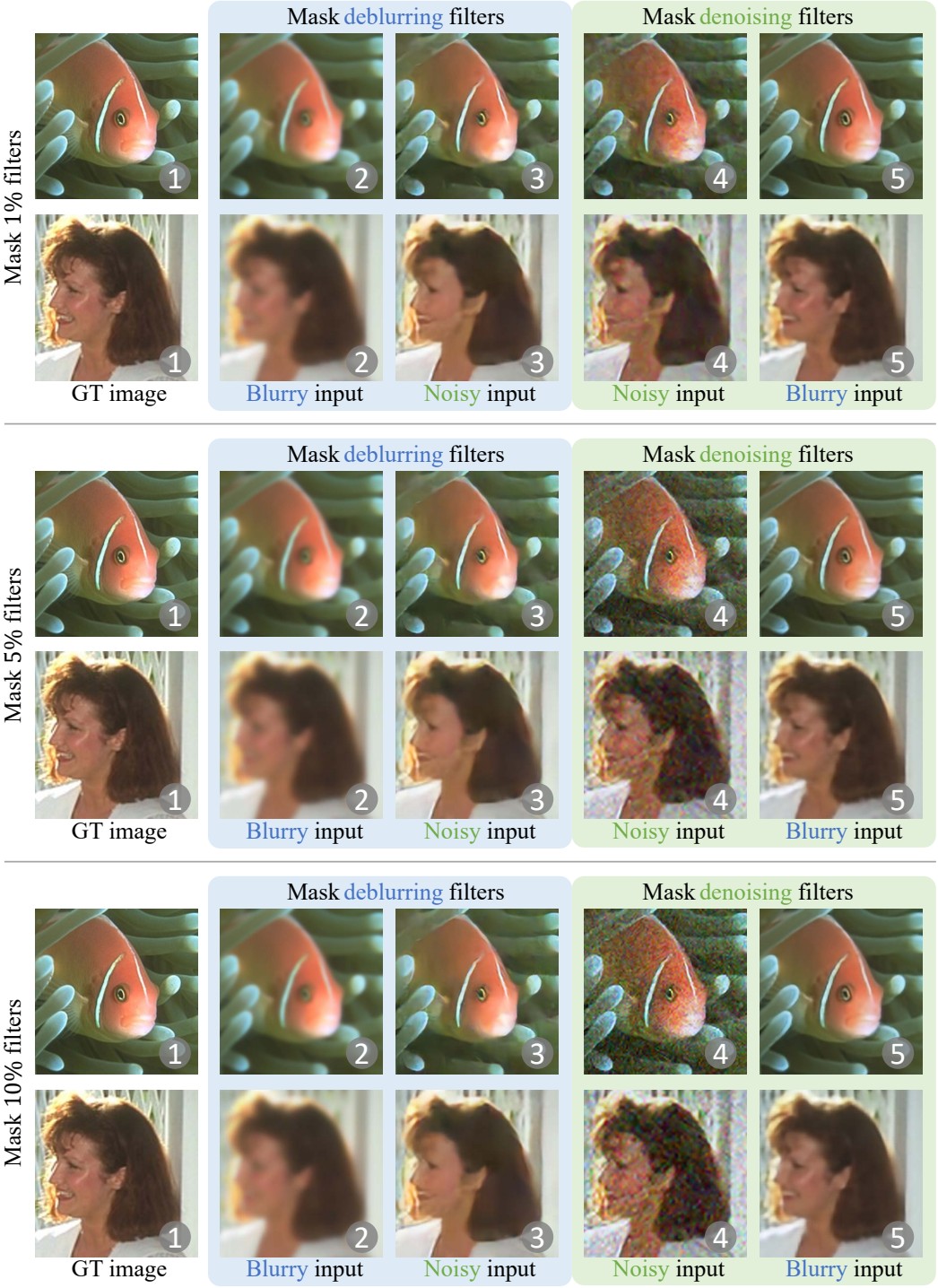

Figure 5: Qualitative results of masking different proportion (from top to bottom: 1%, 5%, and 10%) of discovered filters (by FAIG) in **SRCNN-style** network. Test on **BSD100**. When we mask the deblurring filters, the corresponding network function of deblurring is eliminated (②) while the function of denoising is maintained (③). Similarly, when we mask the denoising filters, the corresponding network function of denoising is eliminated (④) while the function of deblurring is maintained (⑤). Moreover, the performance drop is larger when more discriminative filters are masked. **Zoom in for best view**.

# 3 Mask discovered filters in SRResNet

In the main paper (Fig. 3), we have shown the results of masking discovered filters in SRResNet with different properties. In this supplementary file, we provide more comparison results with other methods: modified IG (*i.e.*, IntInf [2]), absolute values of filter changes ($|\theta - \bar{\theta}|$), and random selection. We measure the MSE error of image gradients as described in Sec. 5.2.1 (in the main paper). A large error/difference represents a large performance drop caused by the replacement of discovered filters.

The comparisons on Set14 [6] and BSD100 [4] are represented in Tab. 5 and Tab. 6, respectively. In both two datasets, our FAIG is able to discover filters that result in larger performance drop, *i.e.*, discover more important filters for corresponding degradations.

Table 5: We compare the performance drop with other methods. For blurry (noisy) inputs, we mask the corresponding debluring (denoising) filters. Larger values indicates a large performance drop, indicating discovering more important/discriminative filters for corresponding degradations. Test on **SRResNet** networks and **Set14**.

| $(10^{-3})$ | mask 1% discovered filters | | | | mask 5% discovered filters | | | |
|---|---|---|---|---|---|---|---|---|
| Input | FAIG (ours) | IG | $\|\theta - \bar{\theta}\|$ | Random | FAIG (ours) | IG | $\|\theta - \bar{\theta}\|$ | Random |
| Blurry image | **6.68**±0.63 | 4.31±1.54 | 0.18±0.13 | 0.07±0.01 | **7.53**±0.24 | 6.41±0.88 | 2.16±0.61 | 0.55±0.32 |
| Noisy image | **6.62**±0.54 | 4.22±0.44 | 0.49±0.10 | 0.04±0.01 | **16.28**±3.84 | 8.01±1.04 | 3.25±1.85 | 0.19±0.05 |

Table 6: We compare the performance drop with other methods. For blurry (noisy) inputs, we mask the corresponding debluring (denoising) filters. Larger values indicates a large performance drop, indicating discovering more important/discriminative filters for corresponding degradations. Test on **SRResNet** networks and **BSD100**.

| $(10^{-3})$ | mask 1% discovered filters | | | | mask 5% discovered filters | | | |
|---|---|---|---|---|---|---|---|---|
| Input | FAIG (ours) | IG | $\|\theta - \bar{\theta}\|$ | Random | FAIG (ours) | IG | $\|\theta - \bar{\theta}\|$ | Random |
| Blurry image | **5.35**±0.42 | 2.43±0.79 | 0.17±0.12 | 0.04±0.01 | **5.98**±0.12 | 4.86±0.73 | 1.60±0.55 | 0.32±0.01 |
| Noisy image | **5.91**±0.92 | 3.82±0.40 | 0.40±0.09 | 0.05±0.01 | **14.06**±2.28 | 7.92±1.17 | 3.01±2.08 | 0.47±0.39 |

**More qualitative results** of masking discovered filters with three different proportions (*i.e.*, 1%, 5%, 10%) are shown in Fig. 6 (on Set14), Fig. 7 (on BSD100).

When we mask the deblurring filters, the corresponding network function of deblurring is eliminated (②) while the function of denoising is maintained (③). Similarly, when we mask the denoising filters, the corresponding network function of denoising is eliminated (④) while the function of deblurring is maintained (⑤). Moreover, performance drop is larger when more discriminative filters are masked. In a word, benefiting from our FAIG, the discovered filters maintain the discriminative property for different degradations.

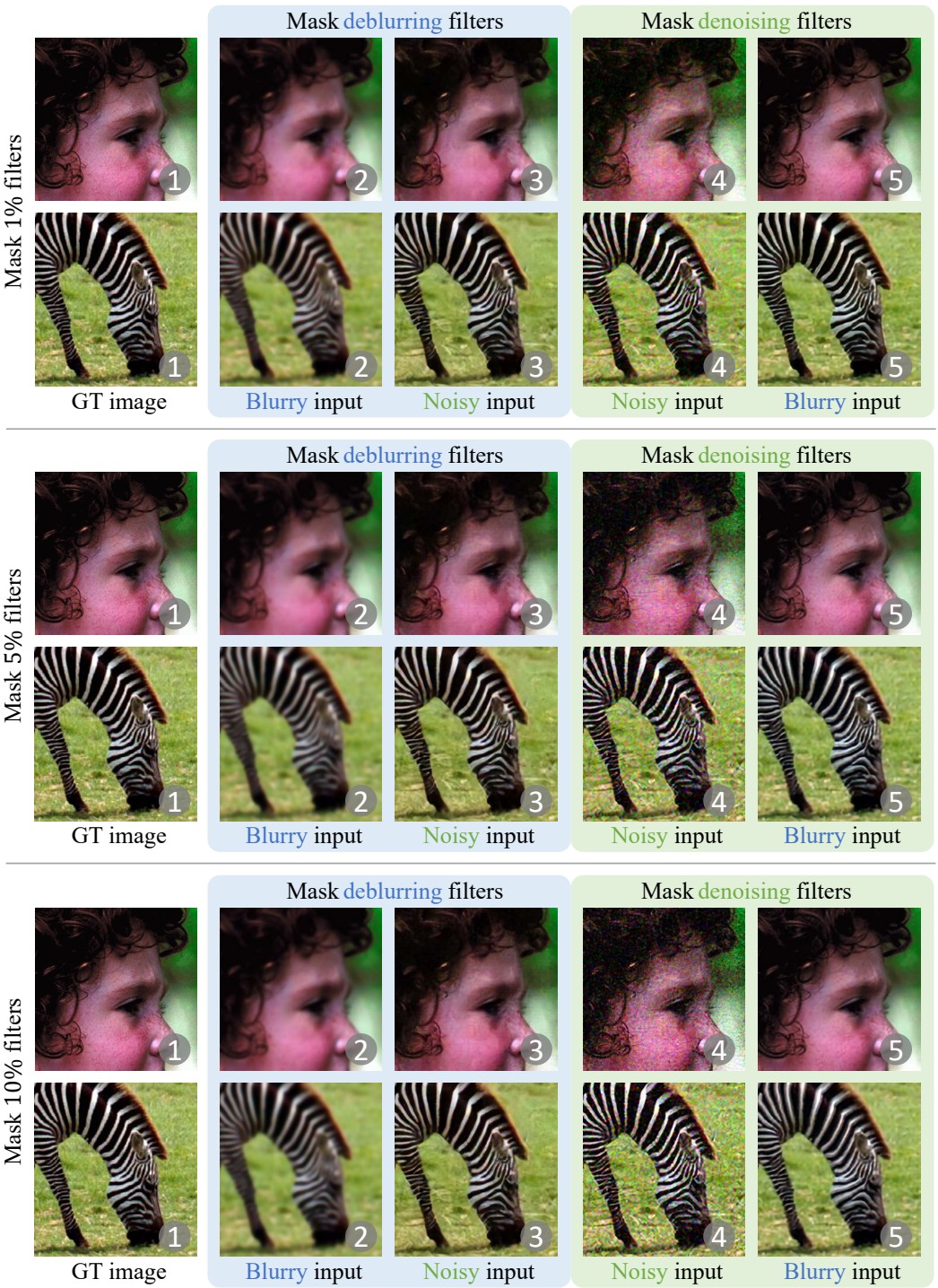

Figure 6: Qualitative results of masking different proportion (from top to bottom: 1%, 5%, and 10%) of discovered filters (by FAIG) in **SRResNet** network. Test on **Set14**. When we mask the deblurring filters, the corresponding network function of deblurring is eliminated (②) while the function of denoising is maintained (③). Similarly, when we mask the denoising filters, the corresponding network function of denoising is eliminated (④) while the function of deblurring is maintained (⑤). Moreover, the performance drop is larger when more discriminative filters are masked. **Zoom in for best view**.

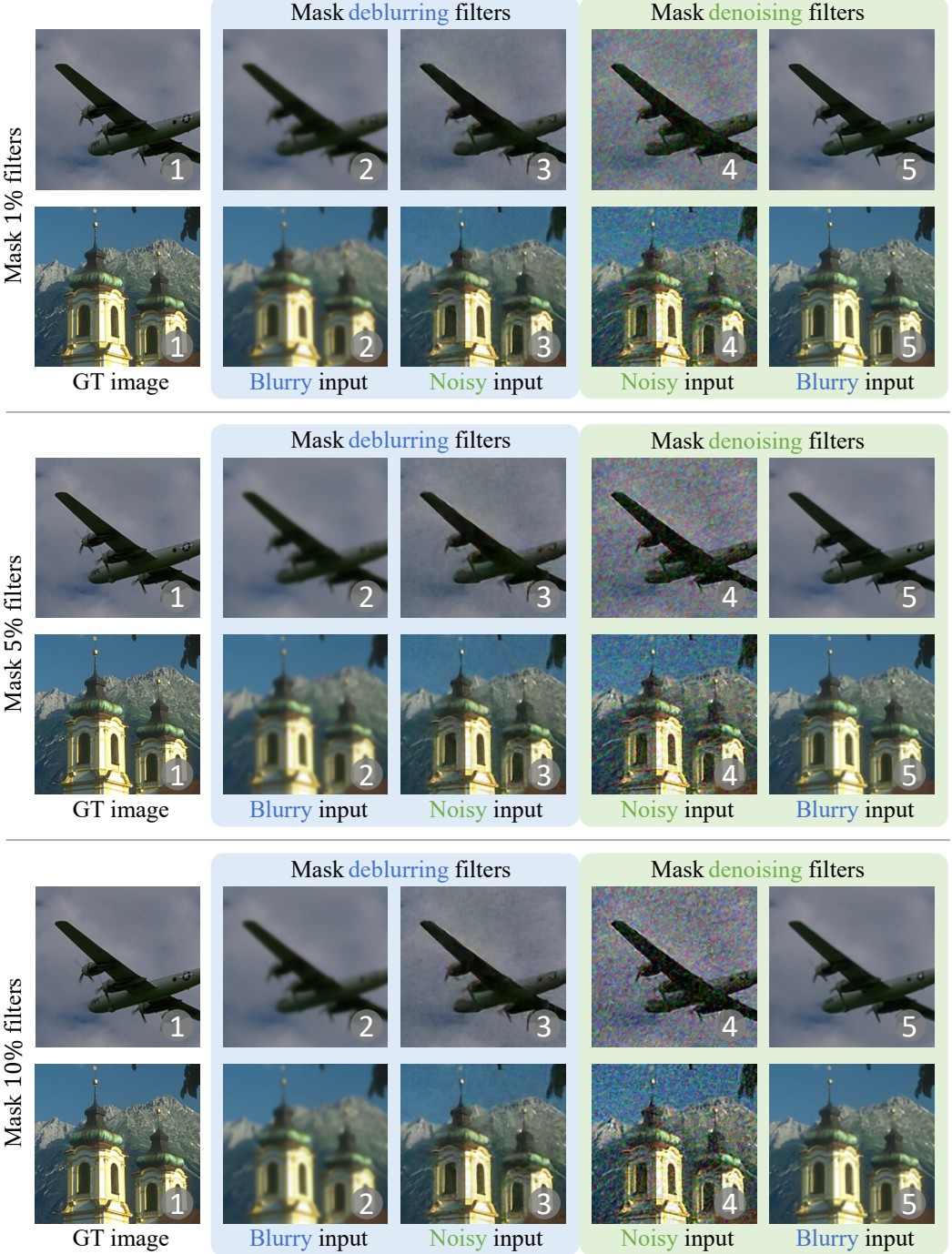

Figure 7: Qualitative results of masking different proportion (from top to bottom: 1%, 5%, and 10%) of discovered filters (by FAIG) in **SRResNet** network. Test on **BSD100**. When we mask the deblurring filters, the corresponding network function of deblurring is eliminated (②) while the function of denoising is maintained (③). Similarly, when we mask the denoising filters, the corresponding network function of denoising is eliminated (④) while the function of deblurring is maintained (⑤). Moreover, the performance drop is larger when more discriminative filters are masked. **Zoom in for best view**.

# 4 Impact of different baseline model

As mentioned in the main paper, we propose a fine-tuning strategy to construct baseline models in FAIG. Specifically, we first train a common SR model for bilinear downsampling kernel as the baseline model $F(\bar{\theta}_{bilinear})$ and then fine-tune it to obtain $F(\theta)$ for bilinear kernel, blur and noise together, as the target model.

The choice of baseline models will affect the final attribution results. We compare our baseline models (denoted as $F(\bar{\theta}_{bilinear})$) with other two alternative baseline models: randomly initialized baseline model $F(\bar{\theta}_{random})$ and all-zero baseline model $F(\bar{\theta}_{zero})$. We conduct the evaluation of masking discovered filters. We mask 1% discriminative filters for corresponding degradations in SRCNN-style network. For a complete comparison, we replace the weights of discovered filters with those filters (at the same location) in $F(\bar{\theta}_{bilinear})$ or their corresponding baseline models, *i.e.*, $F(\bar{\theta}_{random})$ or $F(\bar{\theta}_{zero})$.

The results are shown in Fig. 8. It is observed that: 1) among different baseline models in FAIG (②, ③, ④), our proposed baseline model $F(\bar{\theta}_{bilinear})$ could discovered the most discriminative filters. After masking the corresponding discovered filters, the results of ② (ours) could effectively eliminate the network function of degradation removal. We also show the MSE error of image gradients and our a higher MSE error indicates more important filters we found. 2) If we replace the discovered filters with those filters in $F(\bar{\theta}_{random})$ or $F(\bar{\theta}_{zero})$, the outputs will suffer from severe brightness issues, as the drastic changes of filters badly affect the network outputs. Therefore, our proposed fine-tuning strategy for baseline models is more effective in finding discriminative filters for specific degradations.

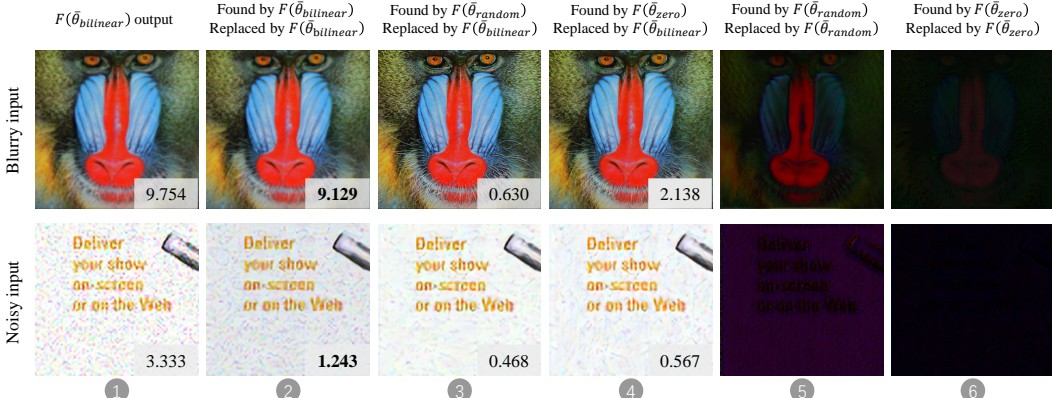

Figure 8: Comparison of masking discovered filters (by FAIG) with different baseline models. We mask 1% discriminative filters for corresponding degradation in SRCNN-style network. "Found by $F(\bar{\theta}_{random})$" means that we discover the important filter with the baseline model of $F(\bar{\theta}_{random})$, while "Replace by $F(\bar{\theta}_{random})$" means that we replace these filters in target model $F(\theta)$ with the filters (at the same locations) in the $F(\bar{\theta}_{random})$ model.

# 5 Degradation prediction

We adopt Set14 to discover filters for a specific degradation $\mathcal{D}$. The thresholds are then calculated on the BSD100 dataset. Finally, the degradation prediction is performed on the DIV2K dataset. We set $\mathsf{T}^{noise}$ and $\mathsf{T}^{blur}$ to 0.6 and 0.5, and the prediction accuracy can reach $98\%$ and $96\%$, respectively. In addition to setting a specific threshold, we also plot the curve of classification accuracy with different threshold values, as shown in Fig. 9. If we set a suitable threshold, we can achieve a high degradation prediction accuracy without any supervision about degradation distinctions.

# 6 Controllable restoration based on the discovered discriminative filters

Based on the discovered discriminative filters for different degradations with our method FAIG, we are able to achieve controllable restoration. Specially, we could interpolate the corresponding

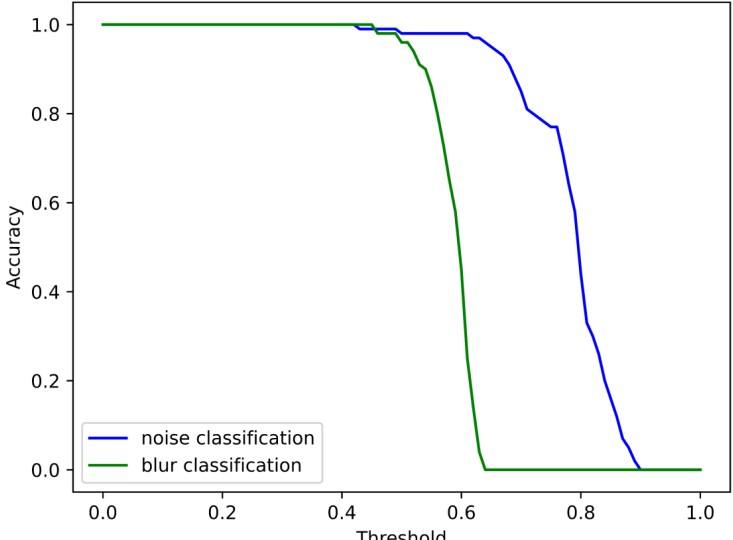

Figure 9: Results of accuracy with different thresholds. Based on the discovered filters, we can predict the degradation of input images by comparing our proposed overlap score (OS) with the threshold.

parameters (at the same location) of discovered filters between the baseline model $F(\bar{\theta})$ and the target model $F(\theta)$ to derive a new interpolated model $F(\theta_{\text{interp}})$, whose parameters are:

$$\theta_{\text{interp}} = (1 - \lambda)\bar{\theta} + \lambda\theta, \tag{1}$$

where $\lambda$ is the interpolation coefficient. As shown in Fig. 10 and Fig. 11, by smoothly adjusting $\lambda$, we can achieve a controllable adjustment of restoration strength without introducing extra parameters. Note that the adjustments of denoising and deburring effects can be realized in one network, by controlling corresponding discriminative filters.

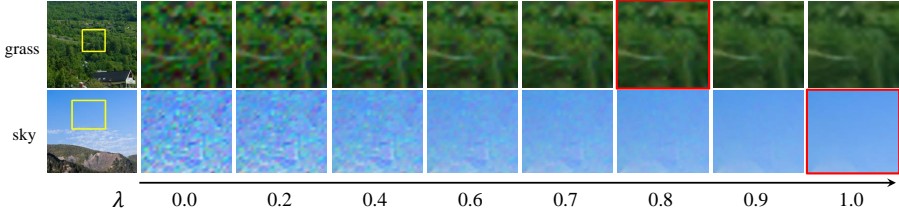

Figure 10: We can adjust the coefficient $\lambda$ for denoising filters to obtain continuous denoising effect. **Zoom in for best view**.

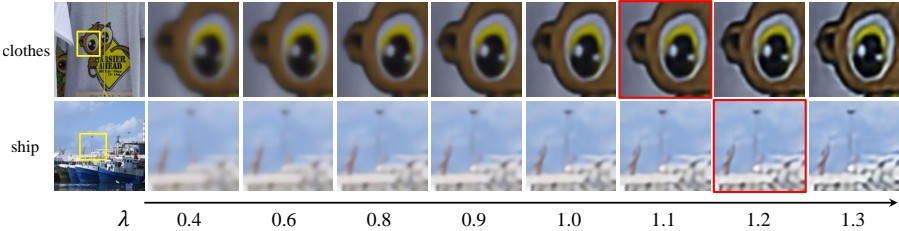

Figure 11: We can adjust the coefficient $\lambda$ for deblurring filters to obtain continuous deblurring effect. **Zoom in for best view**.

# 7 The core codes for FAIG

The core codes of our method $\text{FAIG}_i(\theta, x)$ are provided as follows.

```python
import glob
import os

import cv2
import numpy as np
import torch
from basicsr.models.archs import interpret_arch as interpret_arch

def FAIG(img1, img2, gt_img, baseline_net_path, target_net_path, total_step,
        conv_index):
    """ Filter Attribution Integrated Gradients of a single image

        When finding blurry filters, img1 is a blurry image,
            while img2 is a noisy image.
        When finding noisy filters, img1 is a noisy image,
            while img2 is a blurry image.

    Args:
        img1 (tensor): with the shape (1, 3, H, W)
        img2 (tensor): with the shape (1, 3, H, W)
        gt_img (tensor): with the shape (1, 3, H, W)
        baseline_net_path: path of baseline model
        target_net_path: path of target model
        total_step (int): total steps in the approximation of the integral
        conv_index (list): index of conv layer in srcnn-style like network

    Returns:
        FAIG_img1: FAIG result of img1
    """
    device = torch.device('cuda' if torch.cuda.is_available() else 'cpu')

    baseline_state_dict = torch.load(baseline_net_path)['params_ema']
    target_state_dict = torch.load(target_net_path)['params_ema']

    # calculate the gradient of two images with different degradation
    total_gradient_img1 = 0
    total_gradient_img2 = 0

    # approximate the integral via 100 discrete points uniformly
    # sampled along the straight-line path
    for step in range(0, total_step):
        alpha = step / total_step
        interpolate_net_state_dict = {}
        for key, _ in baseline_state_dict.items():
            # a straight-line path between baseline model and target model
            interpolate_net_state_dict[key] = alpha * baseline_state_dict[
                key] + (1 - alpha) * target_state_dict[key]

        interpolate_net = interpret_arch.srcnn_style(scale=2)
        interpolate_net.eval()
        interpolate_net = interpolate_net.to(device)
        interpolate_net.load_state_dict(interpolate_net_state_dict)

        # for degradation 1
```

```python
        interpolate_net.zero_grad()
        output1 = interpolate_net(img1)
        # measure the distance between the network output and the ground-truth
        # refer to the equation 3 in the main paper
        criterion = torch.nn.MSELoss(reduction='sum')
        loss1 = criterion(gt_img, output1)
        # calculate the gradient of F to each filter
        loss1.backward()
        grad_list_img1 = []
        for idx in conv_index:
            grad = interpolate_net.features[idx].weight.grad
            grad = grad.reshape(-1, 3, 3)
            grad_list_img1.append(grad)
        grad_list_img1 = torch.cat(grad_list_img1, dim=0)
        total_gradient_img1 += grad_list_img1

        # for degradation 2
        interpolate_net.zero_grad()
        output2 = interpolate_net(img2)
        # measure the distance between the network output and the ground-truth
        # refer to the equation 3 in the main paper
        criterion = torch.nn.MSELoss(reduction='sum')
        loss2 = criterion(gt_img, output2)
        # calculate the gradient of F to every filter
        loss2.backward()
        grad_list_img2 = []
        for idx in conv_index:
            grad = interpolate_net.features[idx].weight.grad
            grad = grad.reshape(-1, 3, 3)
            grad_list_img2.append(grad)
        grad_list_img2 = torch.cat(grad_list_img2, dim=0)
        total_gradient_img2 += grad_list_img2

# calculate the diff of filters between the baseline model and target model
diff_list = []
baseline_net = interpret_arch.srcnn_style(scale=2)
baseline_net.eval()
baseline_net = baseline_net.to(device)
baseline_net.load_state_dict(baseline_state_dict)

target_net = interpret_arch.srcnn_style(scale=2)
target_net.eval()
target_net = target_net.to(device)
target_net.load_state_dict(target_state_dict)
for idx in conv_index:
    variation = baseline_net.features[idx].weight.detach(
    ) - target_net.features[idx].weight.detach()
    variation = variation.reshape(-1, 3, 3)
    diff_list.append(variation)
diff_list = torch.cat(diff_list, dim=0)

# multiple the cumulated gradients of img1 with the diff
# refer to equation 6 in the main paper
Single_FAIG_img1 = total_gradient_img1 * diff_list / total_step
Single_FAIG_img1 = torch.sum(
    torch.sum(abs(Single_FAIG_img1), dim=1), dim=1)

# multiple the cumulated gradients of img2 with the diff
# refer to equation 6 in the main paper
```

```python
        Single_FAIG_img2 = total_gradient_img2 * diff_list / total_step
        Single_FAIG_img2 = torch.sum(
            torch.sum(abs(Single_FAIG_img2), dim=1), dim=1)

        # Find discriminative filters for a specific degradation
        # refer to equation 7 in the main paper
        FAIG_img1 = Single_FAIG_img1 - Single_FAIG_img2
        return FAIG_img1.cpu().numpy()

def main():
    device = torch.device('cuda' if torch.cuda.is_available() else 'cpu')

    baseline_model_path = 'model/baseline_model.pth'
    target_model_path = 'model/target_model.pth'

    gt_folder = 'datasets/gt_folder'
    blur_folder = 'datasets/blur_folder'
    noise_folder = 'datasets/noise_folder'
    total_step = 100
    conv_index = [0, 2, 4, 6, 8, 10, 12, 15, 17]

    noise_img_list = sorted(glob.glob(os.path.join(noise_folder, '*')))
    blur_img_list = sorted(glob.glob(os.path.join(blur_folder, '*')))

    FAIG_average_noisy = 0
    # deal noisy imgs
    # average all the gradient difference in a whole dataset
    for img_idx, path in enumerate(noise_img_list):
        imgname = os.path.basename(path)
        noisy_img = cv2.imread(path, cv2.IMREAD_COLOR).astype(
            np.float32) / 255.
        noisy_img = torch.from_numpy(
            np.transpose(noisy_img[:, :, [2, 1, 0]], (2, 0, 1))).float()
        noisy_img = noisy_img.unsqueeze(0).to(device)

        blurry_img = cv2.imread(blur_img_list[img_idx],
                                cv2.IMREAD_COLOR).astype(np.float32) / 255.
        blurry_img = torch.from_numpy(
            np.transpose(blurry_img[:, :, [2, 1, 0]], (2, 0, 1))).float()
        blurry_img = blurry_img.unsqueeze(0).to(device)

        gt_img_path = os.path.join(gt_folder, imgname)
        gt_img = cv2.imread(gt_img_path, cv2.IMREAD_COLOR).astype(
            np.float32) / 255.
        gt_img = torch.from_numpy(
            np.transpose(gt_img[:, :, [2, 1, 0]], (2, 0, 1))).float()
        gt_img = gt_img.unsqueeze(0).to(device)

        # use FAIG for a single image
        FAIG_noisy = FAIG(noisy_img, blurry_img, gt_img, baseline_model_path,
                          target_model_path, total_step, conv_index)

        FAIG_average_noisy += np.array(FAIG_noisy)

    reverse = True
    if reverse:
        sorted_noisy_location = np.argsort(FAIG_average_noisy)[::-1]
    else:
```

```python
        sorted_noisy_location = np.argsort(FAIG_average_noisy)

    save_noisy_filter_txt = os.path.join('noise/FAIG_filter_index.txt')

    # save the filter index to txt file
    np.savetxt(
        save_noisy_filter_txt, sorted_noisy_location, delimiter=',', fmt='%d')

if __name__ == '__main__':
    main()
```