# OpenReview forum: "Finding Discriminative Filters for Specific Degradations in Blind Super-Resolution"
_NeurIPS.cc/2021/Conference — NeurIPS 2021 Spotlight_

### Official Review · Reviewer_Ub7A · 2021-07-06

**Rating:** 8
**Confidence:** 4

**Summary:**

This paper aims to analyze blind SR models and gain a deeper understanding on how a one-branch CNN functions with different degradations. After empirically showing that a one-branch CNN performs on par with two-branch CNNs that are commonly employed in blind SR, the authors propose a filter attribution method based on integral gradients and make important key observations that a small number of discriminative filters ($\sim$1%) can be found for different degradations in one-branch models.

**Limitations And Societal Impact:**

The authors have adequately addressed the limitations of their work in Section 5.5. I agree with the authors that societal impact is N/A as their work focuses on interpreting blind SR models.

**Main Review:**

Pros:
* This paper presents a simple idea for interpreting filters in deep networks that may otherwise be black-box modules.
* Papers focusing on network interpretability are often geared towards high-level vision tasks such as image classification. However, this paper handles blind SR, and with strong empirical evidence that few discriminative filters exist for different degradations (blur and noise), the readers can gain a deeper understanding into how the internal filters are working for image restoration CNNs. I believe the presented analyses will be a great contribution to the low-level vision community.
* The visual results are very impressive and it can be directly observed that discriminative filters exist for different degradations.
* Quantitative comparisons to other analytical methods by masking and re-training 1% filters show that the proposed method based on filter attribution can find more discriminative filters.
* The authors present abundant results (visual results, comparisons, ablation studies, analyses) in the main paper and the supplementary material.
* The paper is clearly written and provides many insights on blind SR models.
* The authors discuss reasonable limitations of their work.

Suggestions/questions:
* For one-branch backbones, network depth (number of residual blocks) is adjusted to match the computational complexity of two-branch networks. Therefore, in Table 1, one-branch models may perform better due to the increased depth. How about increasing the network width / the number of output channels (e.g. 64 channels to 256 channels) to match complexity? Do the authors think results would be similar in this case? It would be more convincing to show results of one-branch models with increased network width as well.
* Urban100 and Manga109 are also widely used testsets in SR. Adding results on these two datasets would make the analysis more complete.
* In Table 3, more discriminative filters can be found for deblurring than denoising with the proposed method compared to the next best method. Any intuitions why?
* The original DAN model is trained iteratively. Is the one-branch DAN also trained iteratively?
* Minor: Fig. 2 could be mentioned in Section 5.2.1 to support its claims.

**Time Spent Reviewing:**

4

---

> ### Author Response · Authors · 2021-08-10
> **Author Response for Official Review by Reviewer Ub7A**
>
> **Q1: How about the performance of the one-branch network after increasing the network width to match the computation budgets?**
>
> **A1**: We add experiments as you suggested. We increase the channel width of the one-branch SRResNet. The number of residual blocks is adjusted to match a similar computation budget with the corresponding two-branch network.
>
> The results in DASR and DAN are shown in Table 1 and Table 2, respectively. We can observe that if we increase network width while decreasing network depth, the performance has a decline. The results are reasonable because it has been widely known that network depth is important to CNN.
>
> Note that the channel number in the DASR SR network is **64**. In DAN, the **main** channel number is **64**. (Some convolution layers also accept a 10-dimension degradation representation; thus the channel number is 74. While it also has the channel attention layer, whose channel number is 4).
>
> Therefore, we set the channel number to 64 in our one-branch experiments for a fair comparison. In this setting, we can observe a comparable performance between one-branch and two-branch networks.
>
> ---
>
> **Table 1**. PSNR (dB), FLOPs and parameters of DASR and its corresponding one-branch SRResNet networks with similar computation budgets.
>
> |     Method    |     # of blocks    |     # of channels    |     PSNR(dB)    |     FLOPs    |     Parameters    |
> |:---:|:---:|:---:|:---:|:---:|:---:|
> |     DASR official    |     - |     64 (SR net)    |     25.116    |     48.27G    |     7.25M    |
> |           SRResNet   one-branch    |     27    |     64    |     25.143    |     48.32G    |     2.33M    |
> |           SRResNet   one-branch    |     16    |     74    |     25.101    |     48.86G    |     2.03M    |
> |           SRResNet   one-branch    |     1    |     128    |     24.816    |     81.67G    |     1.63M    |
>
> ---
>
> **Table 2**. PSNR (dB), FLOPs and parameters of DAN and its corresponding one-branch SRResNet networks with similar computation budgets.
>
> |     Method    |     # of blocks    |     # of channels    |     PSNR(dB)    |     FLOPs    |     Parameters    |
> |---|---|---|---|---|---|
> |     DAN official    |     -    |     64 (main channels)    |     27.341    |     275.36G    |     4.33M    |
> |           SRResNet   one-branch    |     54    |     64    |     27.288    |     77.03G    |     4.32M    |
> |           SRResNet   one-branch    |     16    |     108    |     27.233    |     103.67G    |     4.31M    |
> |           SRResNet   one-branch    |     10    |     128    |     27.201    |     119.93G    |     4.29M    |
> |           SRResNet   one-branch    |     1    |     256    |     26.777    |     324.72G    |     6.50M    |
>
> ---
> ---
>
> **Q2: Add results on Urban100 and Manga109.**
>
> **A2**: Thanks for your suggestions. We also conduct experiments and analyses on Urban100 and Manga109, and draw similar conclusions as those in the main paper. The results are shown in Table 3. We will add them to the supplementary file.
>
> ---
>
> **Table 3**. Results of masking discovered filters. We compare the performance drop with other methods. For blurry (noisy) inputs, we mask the corresponding deblurring (denoising) filters. Larger values indicate a large performance drop. '*Absolute filter changes*' means $\|\theta-\bar\theta\|$.
>
> |    x10^-3       |       |     Mask 1%   discovered filters    |  |  |  |     Mask 5%   discovered filters    |  |  |  |
> |---|---|---|---|---|---|---|---|---|---|
> |     dataset    |     Input    |     FAIG    |     IG    | Absolute filter changes | Random |     FAIG    |     IG    | Absolute filter changes | Random |
> |     Manga109    |     Blurry |     **10.80**    |     4.85    |     0.35    |     0.22    |     **12.02**    |     9.10    |     3.80    |     0.83    |
> |     |     Noisy|     **6.80**    |     5.38    |     0.59    |     0.43    |     **12.75**    |     8.90    |     3.22    |     0.65    |
> |     Urban100    |     Blurry|     **12.96**    |     4.88    |     0.40    |     0.24    |     **14.63**    |     10.86    |     4.62    |     1.20    |
> |     |     Noisy|     **7.11**    |     4.97    |     0.69    |     0.17    |     **12.83**   |     9.10    |     3.21    |     0.38    |
>
> ---
> ---
>
> **Q3: Any intuitions about the phenomena that more discriminative filters can be found for deblurring than denoising with the proposed method? (Table 3 in the main paper)**
>
> **A3**: From Table 3 in the main paper, we can observe that FAIG can achieve higher PSNR values for blurry inputs than noisy inputs. But we think that it does not indicate that FAIG finds more discriminative filters for deblurring. This phenomenon is because the PSNR calculates per-pixel differences, and the deblurring and denoising tasks have different properties for PSNR.
>
> For the deblurring task, if we cannot remove the blur (or we remove partial blur), the output still has a lower PSNR, as most pixels are blurry and thus have a large difference to GT pixels. But for denoising tasks, if we remove partial noise (or even do not remove noise), more pixels have similar values to GT images and thus have a higher PSNR. This explanation is consistent with what we can observe from related literature. In literature, we can observe that the deblurring task usually has large PSNR differences for different methods than the denoising task.
>
> ---
>
> **Q4: Is the one-branch DAN trained iteratively?**
>
> **A4**: No, the one-branch network is trained non-iteratively. Its training strategy is the same as the SRResNet. So, the iterative training scheme is also not essential for blind SR.
>
> ---
>
> **Q5:  Minor: Fig. 2 could be mentioned in Section 5.2.1 to support its claims.**
>
> **A5**: Thanks, we will improve it.

---

> > ### Comment · Reviewer_Ub7A · 2021-08-19
> > **Enjoyed reading the paper!**
> >
> > The authors have addressed all my questions. Thanks for the additional experiments and clear answers!
> > This paper provides deeper insights in interpreting blind SR models and I particularly like its analytical aspect.
> > Great work! I'll stand by my original score of 8.

---

> > > ### Public Comment · ~Diana_Browns1 · 2022-09-19
> > > **Thanks for help!**
> > >
> > > I might want to cause you to notice the chance to request help recorded as a hard copy a paper at https://myassignments-help.com.au/assignment-help-sydney/. This will assist you with saving your time and simultaneously your paper will be of ensured superior grade - without counterfeiting and on time.

---

### Official Review · Reviewer_fvKT · 2021-07-12

**Rating:** 6
**Confidence:** 4

**Summary:**

Most existing blind super-resolution methods use a two-branch network for degradation prediction and conditional restoration. In this paper, authors show that a one-branch network can also automatically learn to distinguish degradations for blind SR with the proposed novel filter attribution method based on integral gradient (FAIG). By using FAIG, they could find 1% filters as the most discriminative filters for each specific degradation in the one-branch blind SR networks. Experimental results show that these selected discriminative filters do have big influence on the corresponding degradation, while having a minor impact on the other degradation. This finding is promising since it may help to design more efficient architectures for blind SR for future work.






**Limitations And Societal Impact:**

Yes, they have addressed the limitation and potential negative societal impact of their work.

**Main Review:**

However, there are still several issues to be addressed.
1. Authors claim that the proposed FAIG method is able to find a small number of filters as the most discriminative filters for degradation removal in blind SR networks. However, only one setting is validated, i.e., the bicubic downsampling with a scale factor of 2, Gaussian blur with a sigma of 2, and Gaussian noise with a level of 0.1. What about the experiments on other settings, like changing the scale factor, the sigma, or the noise level? By only using this same setting for all experiments may not be able to prove that the FAIG would work for various degradations.

2. One important advantage about using the two-branch network for SR is that they provide an opportunity for training a network which can generalize well on images with multiple degradations, even though these degradations are unseen during training. For example, the two listed two-branch methods DAN[27] and DASR[41] directly apply their pretrained networks on some low-resolution images from real world to show their ability for generalization.
How would this finding be helpful for improving the ability for generalization of the one-branch network, especially on those LR images from real world which may be difficult to separate the blurry part and noisy part from the images?

3. In Figure 3 (4), when masking a smaller number of the denoising filters (<20%) for blurry input, it seems that the random is better than the proposed FAIG, is there any particular reason for that?

4. It is not clear how to determine how much proportion of filters should be selected in advance and obtain close performance as the upper bound.


**Time Spent Reviewing:**

2.5 hour

---

> ### Author Response · Authors · 2021-08-10
> **Author Response for Official Review by Reviewer fvKT**
>
> **Q1: What about the experiments on other settings, like changing the scale factor, the sigma, or the noise level?**
>
> **A1**: To confirm the effectiveness of our proposed FAIG, we add more experiment settings with different degradations as suggested by the reviewer. The settings of these experiments are listed in Table 1. All the experiments are conducted on the SRResNet network. We measure the performance by the MSE error of image gradients as described in the main paper. From the results shown in Table 2, we can observe that the proposed FAIG can also work for various degradation settings, and can find more discriminative filters than other methods.
>
> ---
>
> **Table 1**: More experiment settings with different degradation levels and types.
>
> |Experiment No.|Downsample scale|Blur type|Blur level ($\sigma$)|Noise level|
> |-|:-:|:-:|:-:|:-:|
> |1|3|Isotropic|2|0.1|
> |2|2|Isotropic|4|0.1|
> |3|2|Isotropic|2|0.2|
> |4|2|Anisotropic|(2,3)|0.1|
>
> ---
>
> **Table 2**: Results of masking discovered filters. We compare the performance drop with other methods. For blurry (noisy) inputs, we mask the corresponding deblurring (denoising) filters. Larger values indicate a large performance drop. '*Absolute filter changes*' means  \|$\theta-\bar{\theta}$\|.
>
> | x10^-3 |  | Mask 1% discovered filters |  |  |  | Mask 5% discovered filters |  |  |  |
> |:---:|:---:|:---:|:---:|:---:|:---:|:---:|:---:|:---:|:---:|
> | Experiment No. | Input  | FAIG | IG | Absolute filter changes| Random | FAIG | IG | Absolute filter changes  | Random |
> | 1 | Blurry|     **4.00**    |     1.23    |     0.50    |     0.08    |     **5.41**    |     3.49    |     2.37    |     0.40    |
> | | Noisy|     **4.06**    |     4.00    |     2.56    |     1.53    |     **7.18**    |     4.78    |     3.78    |     1.60    |
> | 2 | Blurry|     **5.65**    |     3.09    |     0.29    |     0.13    |     **5.91**    |     5.39    |     2.58    |     0.59    |
> |  | Noisy|     **10.09**    |     4.81    |     0.87    |     0.06    |     **14.71**    |     11.76    |     2.47    |     0.19    |
> | 3 | Blurry|     **7.08**    |     4.53    |     1.85    |     0.14    |     **7.66**    |     7.41    |     4.21    |     0.79    |
> | | Noisy|     **20.19**    |     15.00    |     1.06    |     0.10    |     **44.52**    |     26.83    |     6.47    |     0.59    |
> | 4 | Blurry|     **5.31**    |     2.88    |     0.43    |     0.21    |     **6.41**    |     5.47    |     2.80    |     0.77    |
> |  | Noisy|     **9.67**    |     5.00    |     0.63    |     0.07    |     **15.48**    |     9.25    |     2.42    |     0.22    |
>
> ---
> ---
>
> **Q2: Two-branch networks can generalize well on unseen degradations. How would this finding be helpful for improving the ability for generalization of the one-branch network?**
>
> **A2**: Thanks for pointing out the generalization ability in blind SR. Before answering your question, we first clarify some points.
>
> **1)** How about the generalization ability of one-branch networks compared to two-branch networks? We have conducted experiments to evaluate the generalization ability of DASR and its corresponding one-branch SRResNet on unseen degradations. The results are shown in Table 3. During training, the range of kernel width $\sigma$ is set to [0.2, 4.0]. In testing, we evaluate the unseen kernel width [4.2, 5]. From Table 3, we can observe that the one-branch SRResNet achieves a slightly better performance on these unseen degradations. A similar phenomenon has also been observed in other settings and in the DAN method. Therefore, from our experiments, one-branch networks have comparable generalization ability compared to two-branch designs.
>
> **2)** Our common knowledge is that two-branch designs probably have better generalization ability, as many methods have shown good qualitative results on real-world images. However, this may require careful examination. **i)** With the two-branch design, we first estimate the degradation type and levels for an unseen image. If the input is out of the distribution, the degradation prediction network probably has an inaccurate estimation. Such a mismatch will inevitably result in inferior restoration performance. **ii)** The reason why those real-world images are well restored with the two-branch designs is that they are closer to the training degradation distribution.
>
> We think that more investigations are required for understanding the generalization ability of two-branch networks. Our preliminary experiments have shown that one-branch networks have comparable or even slightly better performance on generalization ability.
>
> **3)** As for the question of improving the ability for generalization of the one-branch network, it is an open question, as we still know little about the working mechanism of CNN for blind SR. However, some recent works [1,2] with one-branch networks have shown superior performance on real-world images, though they are trained on synthetic degradations. We conjecture that the generalization ability relies more on the **data generation process and training strategy**, rather than the network architecture. More investigations should be done in future works, which is a promising direction.
>
> ---
>
> **Table 3**: PSNR (dB) on unseen degradation levels of DASR and its corresponding one-branch SRResNet.
>
> |Gaussian kernel $\sigma$|4.2|4.4|4.6|4.8|5.0| average|
> |:-:|:-:|:-:|:-:|:-:|:-:|:-:|
> |DASR|25.644|25.188|24.757|24.397|24.095|24.816|
> |One-branch SRResNet|25.952|25.307|24.821|24.463|24.182|24.945|
>
> ---
>
> [1] Wang et al., Real-ESRGAN: Training Real-World Blind Super-Resolution with Pure Synthetic Data, arXiv: 2107.10833, 2021.
>
> [2] Zhang et al., Designing a Practical Degradation Model for Deep Blind Image Super-Resolution, ICCV, 2021.
>
> ---
> ---
>
> **Q3: In Figure 3 (4), it seems that the random is better than the proposed FAIG when masking a smaller number of the denoising filters (<20%) for blurry input?**
>
> **A3**: Our proposed FAIG aims to find the discriminative filters for a specific degradation. However, the discovered filters may not be the least relevant neurons for other degradations. In Fig. 3 (4), we mask the FAIG-discovered denoising filters for blurry inputs. Those discovered denoising filters cannot be guaranteed to be the least relevant neurons for the deblur function. So, the phenomenon in Fig. 3 (4) is reasonable.
>
> Note that we suppress gradients to other degradations in FAIG (Eq. 7 in the main paper), *i.e.*, considering the least relevant neurons for other degradations. But these least relevant neurons are considered *among the discriminative filters instead of the whole network filters*. Therefore, the discovered filters cannot be guaranteed to be the globally least relevant neurons for other degradations.
>
> ---
> ---
>
> **Q4: How much proportion of filters should be selected to obtain close performance as the upper bound?**
>
> **A4**: In Fig.3 in the main paper, as we increasingly mask the proportion of discovered filters, the performance has a continuous decline. So, the proportion of selected filters depends on our **selected threshold of performance drop**. If we want a smaller performance drop, we can select a smaller proportion of filters to mask.
>
> Note that when we say that we discover at least 1% discriminative filters for a specific degradation in the network, we do not claim that we can keep only those filters for this specific degradation, but that those filters play the most important role for remove the corresponding degradation.

---

### Official Review · Reviewer_dqh1 · 2021-07-12

**Rating:** 7
**Confidence:** 5

**Summary:**

This paper proposes a Filter Attribution method based on Integral Gradient (FAIG) to analyze the behaviors of blind super-resolution networks on different degradations. It experimentally proves that a one-branch blind SR network is similar to a well-designed two-branch one in the working mechanism. The degradation prediction branch could be automatically learned in a unified one-branch network. These findings are interesting and can help us better understand the behaviors of blind SR networks.

**Limitations And Societal Impact:**

1. The overlap score OS(x, D)  can help in measuring the similarity of the degradation of x and D. However, the degradation space seems to be continuous and there should be an infinite number of different degradations. How do you predict the degradation by calculating OS(x, D)? I mean that there is an infinite number of D, and you can not calculate all of them to choose the largest one.

2. Is it possible to use your findings to help design a better blind SR network? It will be great if you can provide some ideas.

**Main Review:**

The paper is well-written and easy to read.

The finding that a one-branch blind SR network can achieve comparable performance with a well-designed two-branch one is exciting. The proposed FAIG is also interesting, and the adequate experiments also demonstrate the effectiveness of FAIG in discovering discriminative filers.

However, the reason why the one-branch network can be comparable with the two-branch one is still not well explained.  With the help of FAIG, we can know which filters are more important to the specific degradation, but still do not know why they do. It is better to give a further explanation of how the discovered filters work in removing the corresponding degradations.

**Time Spent Reviewing:**

3 hours

---

> ### Author Response · Authors · 2021-08-10
> **Author Response for Official Review by Reviewer dqh1**
>
> **Q1: The reason why the one-branch network can be comparable with the two-branch one is not well explained. How do the discovered filters work in removing the corresponding degradations?**
>
> **A1**: **1)** We attribute the comparable performance between one-branch and two-branch networks to their similar working mechanism, *i.e.*, **i)** the one-branch network is also able to predict/represent different degradations after training (Sec. 5.3 in the main paper); **ii)** different small sub-networks are automatically learned to remove various degradations inside one-branch networks, which is similar to the degradation removal *conditioned on* the predicted degradations in two-branch designs. Such investigations and explanations may not be explicit but provide insights for bridging one-branch and two-branch networks.
>
> **2)** Let us give detailed explanations of why those FAIG-discovered filters are more important to specific degradations. Consider two simple degradations – pepper noise and Gaussian blur. In traditional image processing, they can be effectively removed by *median filters* and *deconvolution operations*, respectively. So, for different degradations, there exist corresponding effective operations/solutions to remove degradations. Using the corresponding filters/operations is required. Exchanging the operations/solutions will not remove the degradations (*e.g.*, median filters cannot remove Gaussian blurs).
>
> The above is our traditional understanding of image processing. In the one-branch ‘black-box’ CNN, we want to find evidence to support the above explanation OR to discover its own working mechanism.
> Through our experiments in the main paper, our FAIG-discovered filters (a set of CNN convolutional filters) are just like the ‘equivalent’ median filter or deconvolution operation in this case. It is natural to understand that those filters are more important to specific degradations, as they mimic our traditional knowledge about restoration. Note that the situation in blind super-resolution is more complicated, as the degradations are more complex and there probably exists semantic processing inside networks.
>
> So, our findings of the division of labor inside one-branch networks indicate that **i)** the working mechanism of blind SR is very similar to our traditional understanding of image processing. **ii)** for a unified network, the division of labor is a better choice rather than ‘mixing’ different functions. Our proposed FAIG provides experimental evidence to support the above explanations. Those findings bridge the one-branch ‘black-box’ CNN restoration with the traditional understanding of image processing, while the latter is the original inspiration of two-branch designs.
>
> ---
>
> **Q2: How to predict the degradation by calculating OS (x, D) in the continuous degradation setting where there is an infinite number of D?**
>
> **A2**: In this work, we explore the preliminary setting where only a single discrete level for each degradation is considered. The current FAIG and OS (overlapping score) are mainly for this discrete setting, and thus should be modified for the continuous degradation setting, which will be investigated as our next step.
>
> We further present the thoughts/options for the continuous degradation setting. In this more practical setting, **i)** FAIG could not be directly applied, as it is impossible to find discriminative filters for the **infinite** degradation levels. Considering that the corresponding solutions/operations for continuous levels in a specific degradation probably present a ‘continuous’ property, *e.g.*, the deconvolution operations to remove two Gaussian blurs with different levels have the same type but different hyper-parameters. (This phenomenon can also be supported by [1], where interpolating network filters for two degradation levels at ends can result in a new network for the middle degradation level). Therefore, one possible solution is that: we first find a set of filters for different degradation types, and further **determine their contributions (percentage) to a specific level**. **ii)** For the method of predicting the degradation, we also need to improve it. First, we can determine the degradation types using our proposed OS (overlapping score). Then, we may determine the degradation levels by calculating the activation percentage or other related statistics inside networks.
>
> Note that this is our conjecture, but we believe it is a reasonable one. We are still exploring it.
>
> [1] [Wang et al., Deep network interpolation for continuous imagery effect transition. In CVPR, 2019]
>
> ---
>
> **Q3: Is it possible to use your findings to help design a better blind SR network?**
>
> **A3**: Yes, it is one of the significances of investigating the working mechanism behind blind SR. We provide two straightforward ideas to improve the design of networks.
>
> **1)** We can use FAIG-discovered filters to guide SR network pruning. As those discovered filters play an important role in removing degradations and they should not be pruned.
>
> **2)** We can design more effective and efficient network structures with the insights from our investigations. For example, SR networks usually have equal width (the channel number) from the input layer to the output layer. However, from the distribution of FAIG-discovered filters in Sec.5.2.3 (Fig. 4) in the main paper, we have observed that the more deblurring filters are located in the back part of the network while denoising filters locate more uniformly. (We conjecture that the deblurring operation requires a larger receptive field for an equivalent ‘deconvolution’ kernel). This phenomenon may provide a motivation to put more network width (channel number) in the back part of the network for a compact design. Therefore, the new-designed network may look like a pyramid in the dimension of network width.
> We believe more valuable designs can be benefited from deeper insights in exploring the working mechanism of blind SR.

---

### Official Review · Reviewer_vM65 · 2021-07-16

**Rating:** 7
**Confidence:** 4

**Summary:**

Existing methods for blind super resolution have recently adopted a design consisting of two branches where one branch is responsible for predicting (or representing) the degradation and the other branch is responsible for reconstruction while conditioning on the degradation information of the first branch. This paper highlights that a single network with one branch can achieve a similar performance as the two-branch approaches and analyses this in detail. More specifically, the authors design a filter attribution method to find most discriminative filters as opposed to most discriminative input pixels or features. This analysis also allows the development of a simple predictor for degradation.

**Limitations And Societal Impact:**

Limitations are clearly described and I don't expect negative societal impact of this work.

**Main Review:**

While early works in upscaling have focussed on fixed degradations, more recent upscaling work that considers the blind setting has helped making super resolution more practical and flexible. However, to my knowledge it has not yet been explored to what extent the solutions deployed in the blind methods are strictly required and in which way a single network can achieve similar performance. As such I find this paper to provide an interesting and fairly unique contribution and I certainly enjoyed reading the paper.

Through the analysis of filter attribution, the method highlights and dissects how certain filters of an otherwise 'black box' network exhibit a given functionality. It is an interesting insight, that even without any form of supervision about degradation distinctions, it can be observed that small sub-networks emerge that are responsible for different degradations.

The paper clearly lists limitations, which is first and foremost that it considers a toy example in blind SR. It only allows a single discrete level for each degradation and the degradations (blur and noise) do not exist at the same time in a given image.

Nevertheless, I think this is an interesting direction to gain more understanding on and in follow up work, it would be interesting to analyse the behaviour for a wider range of continuous degradation settings. In terms of related work, in the space of blind super resolution [CORNILLÈRE et al., Blind Image Super-Resolution with Spatially Variant Degradations, Siggraph Asia 2019] could be cited.

**Time Spent Reviewing:**

2-3 hours

---

> ### Author Response · Authors · 2021-08-10
> **Author Response for Official Review by Reviewer vM65**
>
> Thanks for your acknowledgment of our contributions.
> 1.	Our work is a preliminary investigation of the working mechanism behind blind super-resolution. We believe that our observations and insights will inspire further findings in this field. Further analysis for a wider range of continuous degradation settings will be left as the next-step actions.
> 2.	Thanks for pointing out the missing related work. We will add the citation of [CORNILLÈRE et al., Blind Image Super-Resolution with Spatially Variant Degradations, Siggraph Asia 2019].

---

> > ### Comment · Reviewer_vM65 · 2021-08-30
> > **Thank you**
> >
> > Thanks for addressing my comments. I'm happy to stand by my initial decision to recommend accepting the paper.

---

### Decision · Program_Chairs · 2021-09-27

**Decision:**

Accept (Spotlight)

**Comment:**

All reviewers are enthusiastic about the paper's findings and gave it high scores. The use of one-branch instead of two for blind super resolution is interesting, and one reviewer was particularly interested in the analytical aspect of the paper.